# Additive Manufacturing: A Comprehensive Review

**DOI:** 10.3390/s24092668

**Published:** 2024-04-23

**Authors:** Longfei Zhou, Jenna Miller, Jeremiah Vezza, Maksim Mayster, Muhammad Raffay, Quentin Justice, Zainab Al Tamimi, Gavyn Hansotte, Lavanya Devi Sunkara, Jessica Bernat

**Affiliations:** Department of Biomedical, Industrial and Systems Engineering, School of Engineering and Computing, College of Engineering and Business, Gannon University, Erie, PA 16541, USA

**Keywords:** additive manufacturing, 3D printing, material extrusion, fused deposition modeling, VAT polymerization, binder jetting, material jetting, power bed fusion, CAD, review

## Abstract

Additive manufacturing has revolutionized manufacturing across a spectrum of industries by enabling the production of complex geometries with unparalleled customization and reduced waste. Beginning as a rapid prototyping tool, additive manufacturing has matured into a comprehensive manufacturing solution, embracing a wide range of materials, such as polymers, metals, ceramics, and composites. This paper delves into the workflow of additive manufacturing, encompassing design, modeling, slicing, printing, and post-processing. Various additive manufacturing technologies are explored, including material extrusion, VAT polymerization, material jetting, binder jetting, selective laser sintering, selective laser melting, direct metal laser sintering, electron beam melting, multi-jet fusion, direct energy deposition, carbon fiber reinforced, laminated object manufacturing, and more, discussing their principles, advantages, disadvantages, material compatibilities, applications, and developing trends. Additionally, the future of additive manufacturing is projected, highlighting potential advancements in 3D bioprinting, 3D food printing, large-scale 3D printing, 4D printing, and AI-based additive manufacturing. This comprehensive survey aims to underscore the transformative impact of additive manufacturing on global manufacturing, emphasizing ongoing challenges and the promising horizon of innovations that could further elevate its role in the manufacturing revolution.

## 1. Introduction

Additive manufacturing (AM), widely recognized as 3D printing, has undergone a significant evolution since its inception in the 1980s, transitioning from a rapid prototyping tool to a viable manufacturing method for a broad spectrum of applications. The technology’s advancement is marked by innovations in materials, technology, and software, propelling AM into industries such as aerospace, automotive, healthcare, and fashion [1]. Today, 3D printing is celebrated for its unparalleled customization capabilities, reduced waste, and the ability to fabricate complex geometries that are difficult or impossible to achieve with traditional manufacturing methods [2]. This evolution has transformed AM into a sophisticated array of processes that cater to a diverse range of materials, including polymers, metals, ceramics, and composites, driving innovations across numerous sectors. As AM continues to advance, it is set to revolutionize production processes, supply chains, and product design worldwide, marking its significance as a transformative technology in the manufacturing realm and outlining a future where on-demand customized production can become a reality across industries.

AM distinguishes itself from traditional formative and subtractive methods through its capacity for complex geometries and customization with minimal waste, making it ideal for prototypes and small production runs. Figure 1 shows the comparison of cost per part over different numbers of parts between AM and traditional manufacturing methods [3]. However, AM faces limitations in material variety, part size, and requires significant post-processing, potentially leading to anisotropic mechanical properties. In contrast, traditional manufacturing excels in mass production efficiency, offering superior mechanical strength, isotropy, and surface finish due to a wider range of materials and established processes. While AM can reduce initial costs by eliminating the need for molds or tooling, its cost advantages diminish with scale, unlike traditional methods that benefit from economies of scale. Thus, the choice between AM and traditional methods hinges on the project’s specific demands regarding complexity, volume, and cost efficiency.

The most popular 3D printing technology currently is still undoubtedly the fused deposition modeling (FDM), developed by Stratasys in the late 1980s, is one of the most widely used AM technologies. It works by extruding thermoplastic filaments through a heated nozzle, laying down material layer by layer to construct an object. Recent advancements have focused on enhancing the mechanical properties of FDM-printed parts through the incorporation of reinforced composite filaments and optimizing process parameters for better surface finish and dimensional accuracy [4]. Stereolithography (SLA), patented by 3D Systems in 1986, was the first commercial AM process. It uses an ultraviolet laser to cure photosensitive resins in a layer-by-layer fashion. Technological advancements in SLA have been directed towards improving the speed and resolution of the printing process, enabling the production of parts with high detail and smooth surface finishes [5]. Digital light processing (DLP) technology, similar to SLA, uses a digital light projector to cure photosensitive resins. It stands out for its high printing speed and ability to produce parts with fine details. Advances in DLP technology include the development of new resin formulations that expand the mechanical and thermal properties of printed objects, enabling their use in more demanding applications [1]. Selective laser sintering (SLS) and direct metal laser sintering (DMLS) are powder bed fusion technologies that use lasers to sinter powdered materials, binding them together to form a solid structure. These technologies have seen significant development in the variety of materials that can be processed, including metals, polymers, and ceramics, allowing for the production of functional parts with complex geometries. Recent research has focused on optimizing process parameters and post-processing techniques to improve the mechanical properties and surface quality of printed parts [6]. Material jetting works similarly to inkjet printing, where droplets of material are selectively deposited and cured. Binder jetting involves depositing a liquid binding agent onto layers of powder material. Both technologies have advanced in terms of material diversity and printing resolution. Recent innovations aim at expanding the range of applications by developing new materials with enhanced properties and reducing production costs [7].

The future of AM is poised for growth along several vectors, including the development of new materials, process innovations, and the integration of AM technologies into traditional manufacturing lines. Significant research is focused on scalable AM techniques for large structures, bioprinting for medical applications, and the integration of artificial intelligence to optimize printing processes and material properties [8]. AM continues to push the boundaries of what is possible in manufacturing, driving towards a future where customized production on-demand can become a reality across industries. As the technology matures and overcomes current limitations related to speed, cost, and material properties, its role in the next manufacturing revolution becomes increasingly significant.

Despite the remarkable achievements of 3D printing technology, encompassing a wide range of industries from healthcare to aerospace with its innovative applications, it continues to face significant limitations that highlight its future development potential. One of the primary challenges is the speed of printing, which, despite improvements, remains slow for mass production applications, limiting its utility to prototyping and small-scale manufacturing [9]. Material limitations also pose a significant barrier. The range of materials suitable for 3D printing is expanding, but still lacks the diversity and performance characteristics of those used in traditional manufacturing processes. Furthermore, the resolution and surface finish of printed objects often require extensive post-processing to meet industry standards, adding time and cost to the manufacturing process. Additionally, the cost of 3D printers and materials can be prohibitively high for widespread adoption, particularly in developing countries. There are also challenges related to intellectual property rights and the potential for copyright infringement with the ease of replicating designs [10]. Looking forward, the evolution of 3D printing technology promises advancements in printing speed, material diversity, and cost efficiency. Innovations such as and enhanced precision are on the horizon, aiming to overcome current limitations and expand the technology’s applicability across more sectors, paving the way for a future where 3D printing plays a pivotal role in global manufacturing and beyond.

In this paper, the main workflow of AM is first discussed, including design and modeling, conversion to STL, slicing, 3D printing, and post-processing. Then, different types of AM technologies are introduced, including material extrusion, VAT polymerization, material jetting, binder jetting, selective laser sintering, selective laser melting, direct metal laser sintering, electron beam melting, multi-jet fusion, direct energy deposition, carbon fiber reinforced, laminated object manufacturing, etc. The principle of operation, advantages and disadvantages, materials, main applications, and developing trends of each type of AM technology are discussed in detail. Finally, the challenges, development trends, and potential innovations of AM are analyzed, including important topics such as 3D bioprinting, 3D food printing, large-scale 3D printing, 4D printing, and AI-based AM technologies, as well as new materials, quality control, post-processing innovations, standardization, regulatory, circular economy, and sustainability.

## 2. Additive Manufacturing Process

Additive manufacturing (AM), commonly known as 3D printing, has revolutionized the way objects are created, from simple models to complex structures used in various industries. The process involves adding material layer by layer to create objects from 3D model data, contrary to traditional subtractive manufacturing methods. The overall pipeline of AM encompasses several critical steps, each contributing to the final product’s accuracy, quality, and functionality. Each step in the AM pipeline is vital for ensuring that the final product meets the desired specifications and quality standards. Advances in software, materials, and printing technologies continue to expand the possibilities of what can be achieved with 3D printing, making it an increasingly integral part of modern manufacturing processes Figure 2. The overall workflow of the additive manufacturing process is given below.

### 2.1. Design and Modeling

The design phase marks the beginning of the AM process, where the creation of a 3D model is pivotal. Utilizing Computer-Aided Design (CAD) software tools, designers and engineers can craft detailed and complex digital representations of the desired object. This stage is characterized by meticulous planning of each aspect of the model, ensuring that every curve, edge, and dimension aligns perfectly with the final product’s requirements. A number of critical factors influence the design process.

Design Complexity and Material Considerations: CAD software facilitates the creation of intricate designs that are often impossible to achieve through traditional manufacturing methods. This complexity enables the exploration of new design paradigms and functionalities. Concurrently, the choice of printing material plays a significant role, as different materials offer varying properties, such as durability, flexibility, and thermal resistance. These properties must be aligned with the object’s intended application, dictating specific design adjustments to cater to the chosen material’s strengths and weaknesses.

Technology Limitations and Object Orientation: The capabilities and limitations of the selected 3D printing technology significantly impact the design process. Technologies such as fused deposition modeling (FDM), stereolithography (SLA), and selective laser sintering (SLS) each have unique specifications regarding resolution, accuracy, and the ability to create complex structures. Moreover, the size of the print bed limits the maximum dimensions of the object, potentially necessitating the division of larger objects into smaller, assembleable parts. The orientation of the model during printing also requires careful consideration to optimize the print’s quality, strength, and the necessity for support structures.

Optimization Strategies: Design optimization for AM can take several forms, including topology optimization for material efficiency and structural integrity, and strategic orientation to reduce the need for support structures and improve surface finish. Techniques such as hollowing are employed to minimize material use and print time without compromising the object’s durability.

The design stage, therefore, is not just about creativity but also about a deep understanding of the chosen material, printing technology, and the practical applications of the final product.

### 2.2. Conversion to STL

Once the 3D model is complete, it is exported as an STL file. The conversion of 3D models into stereolithography (STL) files represents a crucial step in the pipeline of 3D printing, serving as the bridge between digital design and physical realization. This process involves transforming a 3D design into a format that 3D printers can interpret and use to build objects layer by layer.

The STL format simplifies the model by breaking down its surface into a series of triangles, also known as tessellation. Each triangle is described by the coordinates of its vertices and the direction of its normal (an outward-facing vector perpendicular to the triangle’s surface). This simplification is crucial because it translates complex geometrical shapes into a uniform language of triangles that 3D printers can understand.

STL files contain many important characteristics such as resolution and manifoldness. The resolution of an STL file depends on the size of the triangles. Smaller triangles result in higher resolution, producing more detailed prints, but at the cost of increasing the file size. Besides, the STL model must also be “watertight”, meaning it should not have any gaps or holes, and all normals should point outward. Non-manifold models can lead to printing errors. After exporting to STL, the file undergoes a slicing process, where specialized software further translates the STL file into G-code. Before slicing, users may adjust the model’s orientation, scale, and other parameters to optimize printing.

There are some potential issues in the STL files. STL files can sometimes contain errors such as reversed normals or non-manifold edges. Many slicing programs include repair functions to automatically fix these issues. To reduce printing time and material use, designers might optimize their models for the STL format by minimizing overhangs and supports.

Converting 3D models into STL files is a fundamental step that transforms intricate digital designs into a language that 3D printers can effectively interpret, setting the stage for the AM process. This conversion encapsulates the transition from conceptual design to physical object, embodying the core of what makes 3D printing a revolutionary manufacturing technique.

### 2.3. Slicing

Once the STL file is generated, the next essential step is slicing, which is a pivotal process where specialized software, often referred to as a slicer, transforms the STL file into a series of thin, horizontal layers. This conversion is critical because it breaks down the complex geometry of the model into a stack of manageable, printable layers, effectively translating the digital design into a physical object. The slicing software plays a multifaceted role in the 3D printing process.

Generation of G-code: The primary output of the slicing process is the generation of G-code, a universally recognized programming language for Computer Numerical Control (CNC) machinery, including 3D printers. G-code is meticulous and detailed, containing precise instructions that direct the printer’s movements, such as the path it should follow, the speed of the extrusion head or laser, and where to start and stop extruding material. This code is what makes it possible for the 3D printer to materialize the digital model into a tangible form, layer by layer.

Customization of Print Parameters: Slicing software offers users the ability to customize various print parameters, which significantly impact the quality, strength, and appearance of the final print. These parameters include the layer height, which determines the print’s resolution; the fill density, affecting the object’s solidity and weight; and the necessity for support structures, which are crucial for printing overhangs and undercuts without deformation. Adjusting the print speed can also influence the print’s quality, with slower speeds generally resulting in higher-quality finishes.

Optimization for Material and Printer: Different materials and printers have unique capabilities and limitations. The slicing stage allows for optimization tailored to the specific characteristics of the material (such as temperature sensitivity, shrinkage, and required post-processing) and the printer’s specifications (such as precision, maximum print size, and compatible materials). This optimization ensures the final print meets the desired criteria for functionality and aesthetics.

Simulation and Troubleshooting: Many slicing programs provide simulation tools that preview the printing process, offering valuable insights into potential issues, such as insufficient supports, overly thin walls, or areas prone to warping. This preview enables users to make informed adjustments before committing to the actual print, saving time, materials, and effort by reducing the likelihood of print failures.

After adjusting these parameters and settings, the slicer recompiles the model into an optimized G-code file, ready to be sent to the printer. This step marks the transition from a digital design to a set of actionable instructions tailored to the unique characteristics of the 3D printer and the material selected. The slicing process, therefore, is not merely a conversion but a critical phase of preparation that bridges digital modeling with the physical act of printing, ensuring that the envisioned design materializes accurately and efficiently in the real world.

### 2.4. Preparation, Printing, and Post-Processing

With the G-code prepared, the stage is set for the 3D printer to initiate the manufacturing process. Preparing a 3D printer for operation involves a series of essential steps to ensure optimal performance and print quality. Initially, the material installation is carried out, which varies depending on the type of printer; FDM printers use a filament that must be loaded into the extruder, while SLA printers require the pouring of liquid resin into the resin tank. Following this, leveling the print bed is crucial to guarantee the proper adhesion of the first layer, a step that may involve manual adjustments or rely on auto-leveling features depending on the printer model. The final preparatory step is pre-heating the printer to the appropriate temperatures for both the print bed and the extruder in the case of FDM printers, ensuring materials are at the optimal temperature for printing. This pre-heating process aids in material adhesion and reduces issues such as warping, thus contributing to the overall quality of the print. These preparatory steps, from material installation to pre-heating, are critical for achieving successful printing outcomes and maintaining the printer’s longevity and safety.

As the printing begins, the printer’s nozzle, or in some technologies, the laser or projector, precisely follows the paths dictated by the G-code. This path guides the deposition of material, constructing the object in a successive layering approach. The choice of material is crucial and is selected based on the object’s intended use, ranging from thermoplastics, photopolymers, and metals to more specialized materials, such as ceramics and composite filaments. Each layer solidifies upon deposition or through subsequent curing, gradually forming the final object with a high degree of accuracy and complexity. More details of different 3D printing technologies are discussed in Section 3.

The completion of the printing phase marks the beginning of another essential phase: post-processing. This stage is vital for enhancing the physical properties and aesthetic qualities of the printed object. Common post-processing methods include the removal of support structures, which are often necessary to prevent the collapse of overhanging features during printing. Surface finishing techniques, such as sanding, polishing, or chemical bathing, are applied to improve smoothness and appearance. For resin-based prints, curing under UV light is a critical step to achieve full strength and stability. Additionally, objects may undergo secondary processes, such as painting, coating, or metal plating, to meet specific functional or aesthetic requirements. The necessity and complexity of post-processing vary significantly with the printing technology used and the end-use of the object. For instance, metal prints might require stress-relief heat treatments and machining to achieve precise tolerances, while plastic prints might need minimal finishing. Despite the advancements in 3D printing technologies, post-processing remains an indispensable step in many cases, affecting the overall time and cost of production. The ongoing development in AM aims to reduce the dependency on extensive post-processing, striving for a future where prints emerge from the printer ready for immediate use, further streamlining the manufacturing process and expanding the applicability of 3D printing across industries.

## 3. Additive Manufacturing Technologies

In this section, different types of AM technologies are discussed, including their principle of operation, advantages and limitations, common materials, main applications, and development trends. Table 1 compares the advantages, limitations, and common materials between these different types of AM technologies. More details are discussed one by one later in this section.

### 3.1. Material Extrusion

Three-dimensional printing has become a more common way to produce materials at a reduced cost, time, and expense. One common type is Material Extrusion (ME), also known as fused deposition modeling (FDM). The main premise of FDM printers involves the creation of parts by extruding thermoplastics layer by layer as opposed to subtractive manufacturing [11].

#### 3.1.1. Principle of Operation of FDM

The production of each printer, in this case, FDM, is unique. Prior to each print, print orientation and the raster design angle must be selected to decrease the strain on the created object [12]. PLA is the typical material deposited to create each object. The structure of typical FDM printers is shown in Figure 3. As printing begins, two different materials are extruded through the extrusion head, i.e., one to create the desired part and the second to create supports for the part. The thermoplastics are heated through the extrusion heads at a temperature between 190 and 230 degrees Celsius and are then printed in strands layer by layer. As they are heated, they turn to liquid but are immediately cooled after being printed and fuse with the layer beneath them [13]. As the target part is made, the support material and part material are alternated between per layer. The initial layer begins on the base plate and is then built upon as the material deposited onto each layer [14]. Since the build is three-dimensional, the x, y, and z dimensions each have a motor to control movement on each axis. This leads to a bottom-up approach in FDM. The completed part can be removed from the printer by holding the part and twisting the tray. This will release the part and the supports can be broken or cut off.

#### 3.1.2. Advantages and Limitations of FDM

The main advantages of FDM printing are the low price, their ability to be reproduced, their speed efficiency, and low maintenance [16]. However, there are also many different limitations facing FDM printers. This type of printer typically requires a higher temperature and supports for angles greater than 45 degrees [17]. These supports lead to materials being wasted and an increased production time. One of the biggest limitations of FDM is its inability to produce a part regardless of the geometry. These limitations will be discussed in a later section of this review.

#### 3.1.3. Materials of FDM

FDM primarily employs thermoplastics for their advantageous properties of being easily heated and reshaped, catering to a wide range of applications. Among the materials commonly utilized in FDM, polylactic acid (PLA) stands out for its unique combination of being a thermoplastic that is also biodegradable, making it an environmentally friendly option. This characteristic, coupled with its safety for use in sensitive environments such as hospitals and food packaging, underscores its popularity in FDM applications. Besides PLA, FDM technology often uses materials such as acrylonitrile butadiene styrene (ABS), polyethylene terephthalate glycol (PETG), and various polyolefins [18]. These materials are chosen for their robustness and flexibility, although they initially present more significant mechanical constraints compared to PLA. These limitations can affect the durability and functional applications of printed objects, necessitating careful consideration in material selection based on the specific requirements of the intended application.

High-performance thermoplastics such as polyether ether ketone (PEEK) and polyetherimide (PEI) are valued for their exceptional mechanical properties and resistance to high temperatures. These materials are essential in applications demanding durability, chemical resistance, and thermal stability, such as in aerospace, automotive, and medical industries. However, their use in FDM requires specialized printing equipment capable of achieving and maintaining high extrusion temperatures—typically above 350 °C for PEEK and around 340 °C to 360 °C for Ultem. Additionally, a heated print bed and chamber are crucial to ensure proper adhesion and minimize warping, as these materials are prone to contraction upon cooling. The high processing temperatures also necessitate advanced cooling systems and thermal management strategies to prevent material degradation and ensure dimensional accuracy.

#### 3.1.4. Applications of FDM

FDM technology finds application across a broad spectrum of industries, significantly impacting fields from healthcare to automotive and even consumer goods. Within the medical sector, FDM plays a crucial role in enhancing surgical procedures and medical training by facilitating the creation of precise anatomical models. These models are invaluable for surgical preparation and educational purposes, offering a high degree of anatomical accuracy [19]. Furthermore, FDM’s versatility allows for its use in developing specialized medical devices, such as dialysis catheters, innovative drug delivery systems, and scaffolds for tissue engineering, which are pivotal in regenerative medicine practices [20,21,22]. In the automotive industry, FDM’s contribution extends to the production of durable and lightweight components such as window car holders, showcasing the technology’s ability to produce parts with complex geometries and tailored mechanical properties at a reduced cost and turnaround time [23]. Beyond professional applications, FDM’s accessibility and affordability have made it a favorite among hobbyists and educators. It serves as a tool for recreational projects, educational models, and prototype development, underlining its versatility and the broad appeal of 3D printing technologies. This wide-ranging utility of FDM underscores its transformative potential across various sectors, from revolutionizing medical practices to enhancing manufacturing processes and promoting innovation at the consumer level.

#### 3.1.5. Developing Trends of FDM

As AM becomes more common, advancements are being made. Specifically, FDM has begun to implement five-axis 3D printers. This reduces overhanging structures and supports needed, which overall reduces the material and cost for each print. The two additional axes allow for a rotation during printing. This ensures a more detailed print with less required support [24]. Since AM is increasing in popularity advancements are needed to keep costs low and increase efficiency. Fused deposition modeling is a reliable method with a variety of uses and is a feasible solution to the production of a variety of products. Although there are many types of 3D printing, FDM should not be overlooked.

The innovation of filament core reinforcement with glass, carbon, or basalt fibers in FDM significantly enhances the mechanical properties of printed parts. These fibers, embedded within the core of thermoplastic filaments, provide improved strength, stiffness, and thermal stability compared to standard thermoplastics. Carbon fiber-reinforced filaments, for instance, offer superior strength-to-weight ratios and dimensional stability, making them ideal for aerospace, automotive, and industrial applications. Glass fiber reinforcement improves impact resistance and tensile strength, whereas basalt fibers contribute to excellent thermal and chemical resistance.

Simultaneously, nanotechnology trends in FDM are pushing the boundaries of material performance further. The incorporation of nanoparticles into filaments, such as carbon nanotubes or graphene, enhances electrical conductivity, heat dissipation, and mechanical properties at the nanoscale, opening new avenues for functional and structural applications. These nanocomposites can be tailored for specific uses, ranging from wearable electronics with conductive properties to components requiring enhanced thermal management or mechanical performance.

Both fiber reinforcement and nanotechnology trends in FDM represent a leap towards manufacturing more durable, functional, and customized 3D printed products, bridging the gap between prototyping and end-use manufacturing.

### 3.2. VAT Polymerization

Generally, the VAT polymerization (VP) technique uses a light source to cure a photopolymer resin contained within a reservoir [25]. There are three types of VP—stereolithography, digital light processing, and continuous digital light processing.

#### 3.2.1. Principal of Operation of VPs

Of these three forms of VP, stereolithography (SLA) is distinct from the two similar techniques of regular DLP and continuous DLP (cDLP). Therefore, SLA will be discussed first, followed by both DLPs. Stereolithography utilizes a laser to cure a photopolymer resin onto a build plate in the desired geometry. The laser traces the resin within the reservoir as required per the guidance of a mirror. SLA printers are either top-down or bottom-up oriented. For top-down printers, the build plate starts at the top of the reservoir, the laser cures the layer as specified by the geometry code, and then the build plate moves down corresponding to one increment of layer thickness; thus, each successive layer is above the previous. On the other hand, the build plate of bottom-up printers starts at the bottom of the resin reservoir. Between each curing, the build plate moves up one thickness layer, resulting in each successive layer being below the prior [26]. Parameters that influence which printer orientation is used to print an object includes product size and complexity, time, reliability, etc. Digital light processing is similar to SLA in that a photopolymer resin is being cured; however, this method utilizes a digital light projector for curing. This digital light projector cures the complete layer simultaneously rather than tracing. Because the curer is a digital source, the print will exhibit square-like subunits composing each layer; these are square light pixels called “voxels” [27]. Despite these differences, DLP printers can also be oriented top-down or bottom-up with the same operation. The distinction between DLP and cDLP is the movement of the build plate along the z-axis. For DLP, the build plate moves one layer thickness per cure, while the cDLP build plate continuously moves along the z-axis to accommodate continuous curing, which results in shorter build times [28]. Figure 4 shows the schematic of the SLA 3D printing process and a print sample. Note that a schematic for cDLP is not included because the difference in build plate motion is not visible in an image. Furthermore, two images are provided for SLA to display top-down versus bottom-up printer orientation, but this is applicable to DLP as well. Figure 5 shows the different lighting technologies between SLA and DLP. The can be seen that a laser cures the resin by tracing a path in SLA while a voxel grid is illuminated to cure resin a full layer at a time.

#### 3.2.2. Advantages and Limitations of VP

VAT polymerization, which encompasses technologies such as SLA and DLP, offers several significant advantages in the realm of AM. One of its primary benefits is the ability to produce parts with exceptional detail and surface finish, surpassing most other 3D printing technologies. This precision makes it an ideal choice for applications requiring intricate features, such as jewelry, dental applications, and highly detailed prototypes. Moreover, the variety of resins available for VAT polymerization technologies enables the creation of parts with specific mechanical, thermal, and optical properties. This versatility supports a wide range of applications, from functional prototypes to end-use parts in various industries. The technology also allows for relatively fast printing speeds, especially in the case of DLP, which can cure entire layers simultaneously [30]. Additionally, the support structures required for overhanging parts are generally less extensive than those needed for extrusion-based processes, simplifying post-processing steps.

Despite its advantages, VAT polymerization also has several limitations that can affect its applicability for certain projects. One of the primary drawbacks is the limited build volume, which can restrict the size of parts that can be produced. This makes it less suitable for applications requiring large components or high-volume production runs. Additionally, the materials used in VAT polymerization, typically photopolymer resins, can be more expensive than those used in other forms of 3D printing, such as filament for FDM. The resins can also be brittle and may not possess the same strength or durability as materials used in other manufacturing processes, limiting their use in functional parts that require high mechanical strength [31]. Furthermore, the parts produced by VAT polymerization may suffer from shrinkage and warping due to the curing process, which can affect dimensional accuracy and require additional post-processing to achieve the desired tolerances. Lastly, working with photopolymer resins requires careful handling due to their potential toxicity and the need for proper disposal, posing environmental and safety concerns.

#### 3.2.3. Materials of VP

Most commonly, VAT polymerization is used with a photocurable resin. These resins can range from standard resins with average mechanical properties for general prototyping, to structural, tough and durable, or elastic resins depending on usage or application. Due to the variability of polymers, resins come in many color options, as opposed to other printing materials, such as metals. More advanced materials that can also be used include ceramics or waxes, biocompatible resins, and bioinks; these are typically seen in biomedical applications [32]. Generally, as material quality/complexity increases, so does processing time and costs (both raw materials and equipment).

#### 3.2.4. Applications of VP

Due to the versatility of 3D printing, there are endless applications in which VAT polymerization can be applied. Both SLA and DLP are utilized over a wide range of fields including manufacturing, research development, and medicine, to name a few. In research and development, these methods can quickly create products to test form, fit, and function or bring vision to a project through prototyping without a lot of time or financial investment [33]. Minimal time and financial obligations are also an incentive for manufacturers, particularly in areas of investment castings for metallurgy, because complex, dimensionally accurate molds and cores can be developed, which were otherwise made less accurately through hand carvings or FDM [34]. The revolution of 3D printing and market demand has also made utilizing this equipment for personal entertainment possible.

#### 3.2.5. Developing Trends of VP

One of the greatest selling points to a solution in healthcare is its ability to impact a large range of patients. Because each human is unique, the more customizable a solution, the greater chances for success for a given patient. This makes VAT polymerization a great method for creating medical products. Recently, VAT (more specifically SLA) has been under development for its use and effectiveness with pharmaceuticals. Due to its high accuracy, resolution, speed, and materials, this method is thought to be up and coming for development of micro-scale, yet high concentration drug delivery systems in areas such as pain or disease management [35].

Besides, the biocompatibility of materials used in VP is a critical consideration, especially for applications in the medical and dental fields, where direct or indirect contact with the human body is frequent. Biocompatible photopolymer resins have been developed to meet stringent regulatory standards for medical devices and implants. These materials undergo rigorous testing to ensure they do not elicit an adverse reaction from biological tissue. This includes assessments for cytotoxicity, irritation, sensitization, and, in some cases, longer-term biostability when intended for implantation [36]. The advantage of using VAT polymerization for producing biocompatible parts lies in its ability to create complex, high-resolution structures that can match the specific anatomical features of patients, essential for custom implants, dental restorations, and surgical guides. Furthermore, recent advancements have led to the development of resins that not only meet biocompatibility requirements but also possess properties such as improved mechanical strength, flexibility, and thermal stability, expanding their use in creating functional medical devices [37].

It is also worth noting that the biocompatibility of a printed part is not solely dependent on the material, but also on the printing process and post-processing steps. Proper cleaning and curing are essential to remove uncured resin and achieve the desired material properties, including biocompatibility. In some cases, additional sterilization is required before clinical use. As technology and materials evolve, the range of biocompatible resins and their applications in healthcare continues to expand, offering promising opportunities for personalized medicine and advanced medical treatments.

### 3.3. Material Jetting

Material Jetting (MJ) combines the precision of resin 3D printing with the speed of filament printing, producing parts with realistic colors and textures, primarily aimed at professionals in industries such as automaking, design, healthcare, and product manufacturing. Two major brands, Stratasys and 3D Systems, dominate the production of MJ devices. This technology now extends to diverse applications, from biocompatible dental molds to high-speed manufacturing tooling. Three primary MJ types include PolyJet by Stratasys, Nano Functional Technology using solid nanoparticles, and Drop on Demand (DOD) for thick fluid materials, particularly useful in creating wax models for investment casting in jewelry manufacturing.

#### 3.3.1. Principle of Operation of MJ

Material Jetting (MJ) operates akin to a two-dimensional inkjet printer, but instead of ink, it sprays material onto the construction platform using either a continuous or DOD approach. An illustration of how the Stratasys J55 (Stratasys, Ltd., Eden Prairie, MN, USA; Rehovot, Israel) dispenses print material is shown in Figure 6 [38]. The process involves layer-by-layer construction as the material solidifies upon the substrate, and the print head, equipped with nozzles, moves horizontally across the build platform. The complexity of these machines varies, affecting the deposition methods and the materials used. The limited material options include polymers and waxes due to their viscosity and suitability for forming droplets.

The core components of an MJ system comprise the print head, responsible for precise material deposition; the build platform, where the object gradually forms layer by layer; and a UV or heat source to initiate material solidification. This layer-by-layer printing process allows for intricate shapes and structures. Support structures may be generated as needed and removed post-printing. After completion, post-processing steps might be necessary, including support removal, surface finishing, and additional curing for certain materials. MJ ensures high-resolution and precision in printing, often accompanied by quality control measures to validate the final product against specifications.

In MJ’s operation, the material is heated to an optimal viscosity before tiny beads of photopolymer are deposited by the print head. UV light solidifies the material, forming each layer as the build platform gradually descends. Unlike many other 3D printing methods, MJ employs a line-wise material deposition system, enabling multiple heads to allocate different materials for multi-material printing or dispensing soluble support structures. Despite similarities to SLA in utilizing photo-polymerization, MJ-printed parts achieve desired properties without additional post-curing due to the minute layer thickness employed.

#### 3.3.2. Advantages and Limitations of MJ

Material jetting stands out for its exceptional resolution and intricate detailing, apt for crafting complex geometric shapes with fine features. Capable of achieving layer thicknesses as minimal as 16 microns, it delivers remarkably accurate and smooth surfaces. Additionally, the technology enables multi-material printing, allowing simultaneous use of various materials, encompassing different colors, rigid and flexible materials, or materials with differing properties. This versatility facilitates the production of parts with diverse characteristics within a single print job. Certain material jetting systems excel in full-color printing, ideal for crafting vibrant prototypes, architectural models, and consumer products, enhancing realism in aesthetics. Known for its relatively high speed, particularly in producing small and intricate parts, material jetting serves as a favorable option for rapid prototyping and small-batch production. Moreover, it minimizes or eliminates the need for extensive support structures, as support materials can be easily removed during post-processing, reducing laborious support removal tasks. Embracing a wide array of compatible materials, from photopolymers to thermoplastics, ceramics, and even select metals, material jetting broadens its potential applications across various industries.

Material Jetting encounters several limitations, starting with its high costs, encompassing substantial initial investments and ongoing operational expenses. Moreover, the materials employed can be particularly pricey, especially for full-color printing applications. Another constraint lies in the restricted build volume of material jetting machines, rendering them inadequate for large-scale production or crafting sizable parts. Post-processing becomes a requisite, involving the removal of support materials, surface refinement through sanding, and sometimes the application of additional coatings, consequently elongating production times. Material considerations pose a challenge as well, given the technology’s limitation to specific compatible materials, despite an expanding range, potentially falling short for certain applications. Additionally, the trade-off between resolution and speed presents a hurdle, as achieving high resolution often demands slower print speeds, posing a challenge in striking a balance between these parameters. Material waste emerges as another concern, particularly in discarding support materials, leading to increased expenses and environmental concerns. Lastly, the complexity involved in operating and maintaining material jetting machines necessitates skilled personnel and regular upkeep, adding to the operational intricacies.

#### 3.3.3. Materials of MJ

In material jetting, a diverse range of materials have been utilized, including plastics, polymers, and, intriguingly, some metals. Each material boasts unique properties, significantly influencing both the printing process and the final attributes of the printed object. Typically, these materials exist in fluid or semi-fluid states, easily dispensed through a print head. Commonly used materials in material jetting can be categorized into several types, each demonstrating distinct impacts on the printing process [39].

Polymers represent a substantial portion of material choices in material jetting. Among them, photopolymers reign as the most prevalent. These liquid resins solidify upon exposure to ultraviolet (UV) light, recognized for their remarkable high resolution and capacity for intricate detailing, making them ideal for producing complex objects. Additionally, certain material jetting systems utilize thermoplastic polymers like PLA, ABS, and PETG. These materials, heated and extruded through the print head, enable layer-by-layer deposition and solidification.

Metal material jetting, while less common, stands as a more advanced process. It involves jetting metal powders encased in a polymer matrix or liquid binder. Subsequently, parts undergo sintering or debonding processes to eliminate the binder and fuse the metal particles, resulting in solid metal parts. This method serves applications necessitating high strength and metallic properties.

Ceramic material jetting parallels metal material jetting in approach. Ceramic powders, mixed with a binder, create a printable slurry. Post-printing, the binder is removed, and the ceramic part is often sintered at elevated temperatures to attain desired mechanical and thermal properties. Such material jetting is employed in scenarios requiring high-temperature resistance and electrical insulation.

Composite materials are also compatible with material jetting, involving the mixing of two or more materials during printing. For instance, carbon-fiber-reinforced polymers yield lightweight, robust parts with enhanced mechanical properties.

The selection of materials impacts the printing process in multifaceted ways. First, diverse materials possess distinct optimal printing parameters—such as print speed, layer height, and curing times—necessitating printer configuration tailored to each material. Second, material choice affects the achievable resolution and detailing capabilities, with photopolymers excelling in intricate object production. Third, the mechanical properties, including strength and thermal resistance, vary among materials, requiring alignment with the desired final product properties. Furthermore, post-processing steps—such as sintering, curing, or debonding—may be material-dependent, impacting the complexity and cost of finalizing the product. Finally, specific application needs, such as electrical conductivity, mandate careful material selection tailored to meet these requirements.

#### 3.3.4. Applications of MJ

Material jetting has become a versatile solution applied across various real-world domains due to its capacity for producing high-resolution, multi-material, and multi-color components. Its practical utilization spans diverse industries. For instance, in the domain of prototyping and product development, material jetting plays a pivotal role, especially within sectors like automotive, aerospace, and consumer electronics. Engineers and designers harness this technology for swift prototyping, enabling rapid iterations and efficient design testing before moving into large-scale production. Similarly, in the sphere of dental and medical devices, material jetting showcases its significance by crafting precise and tailored models, orthodontic devices, hearing aids, and surgical guides. Its ability to create intricate, patient-specific parts proves indispensable in these critical applications.

The aerospace industry heavily relies on material jetting to fabricate lightweight, high-performance components such as air ducts, interior panels, and even small satellite parts. Architects and construction professionals leverage material jetting to develop intricate architectural models and prototypes, aiding in effective visualization and communication of design concepts. Additionally, the jewelry and fashion sectors benefit from the technology, utilizing its capabilities to produce finely detailed accessories and jewelry, often integrating a myriad of colors and materials.

Moreover, high-end consumer electronics, including custom smartphone cases, tap into material jetting for its precision and capacity to amalgamate various materials within a single print. Educational institutions and research labs also find value in material jetting, employing it for both educational purposes and research exploration into 3D printing technology and its expansive potential. Furthermore, the shoe and orthopedic industries deploy material jetting to manufacture custom insoles and orthotic inserts, precisely conforming to the wearer’s foot shape for superior comfort and support.

An emerging frontier, 3D food printing recognizes the potential of material jetting in creating intricate and decorative food items or customizing food textures. This indicates its prospective utility in culinary and confectionery applications, potentially revolutionizing the way food is presented and experienced.

#### 3.3.5. Developing Trends of MJ

Material jetting stands as an established 3D printing technology, continuously evolving through ongoing developments and applications. Its growth likely involves embracing new innovations and trends that further enhance its capabilities. Notable advancements within material jetting encompass a range of significant developments.

Firstly, the exploration of hybrid systems integrates MJ with other 3D printing technologies such as FDM and SLA. This approach aims to amalgamate the strengths of multiple technologies within a single print job, potentially unlocking new possibilities and diversifying the applications of material jetting.

Secondly, the technology’s inherent strengths lie in its high resolution and precision. Material jetting excels in creating highly precise 3D-printed objects by jetting tiny droplets of photopolymer material, cured subsequently with UV light. Such precision makes it an ideal choice for applications requiring intricate details and fine surface finishes, particularly evident in industries such as healthcare and prototyping.

Lastly, the influence of MJ expands into multi-material printing capabilities. This unique feature allows for the simultaneous use of diverse materials within a single print job, offering substantial value in crafting objects with varying properties, colors, and functionalities [40]. Industries such as automotive, aerospace, and healthcare benefit immensely from this versatile capacity, enabling the creation of complex, multi-functional components. Additionally, ongoing advancements in material properties, along with a focus on sustainability, further drive the technology’s evolution toward broader applications and more eco-friendly practices.

### 3.4. Binder Jetting

Binder jetting (BJ) was developed at MIT in the 1990s. The printing process of BJ consists of a binder being printed onto a bed of powder to form the cross sections of a print, similar to powder bed fusion printing [41].

#### 3.4.1. Principle of Operation of BJ

The BJ technology employs a precise and controlled process where a printer nozzle systematically traverses a powder bed, depositing a liquid binder to form the desired cross-sectional shape for each successive layer. Following the application of the binder, the system lays down an additional layer of powder. This ensures the freshly applied binder acts as an adhesive, effectively bonding the powder particles to form a solid layer [42]. The size of the liquid binder droplets is meticulously controlled, typically less than 100 microns in diameter, to ensure high precision and detail in the final product. After the binder is applied, the entire print bed undergoes a curing process, either under a heat lamp or within a furnace. This step is crucial for activating the binder and solidifying the powder, thereby imbuing the emerging object with the necessary mechanical strength [43]. Following this, the process of powder layering and binder application is repeated, layer by layer, until the object is fully formed. The interim product of this process, known as the "green body", signifies the nascent stage of the printed object. It has achieved its geometric specifications but requires further post-processing to enhance its structural integrity and surface finish. Through this intricate layer-by-layer approach, binder jetting technology facilitates the creation of complex parts with a high degree of accuracy and detail. One of the most common BJ technologies is ColorJet Printing (CJP). Figure 7 shows the principle of operation of the BJ technology [44].

#### 3.4.2. Advantages and Limitations of BJ

The BJ technology offers several compelling advantages compared to alternative AM techniques, making it an attractive option for a range of applications. Notably, BJ is recognized for its cost-effectiveness, which stems from the efficient use of materials and the process’s scalability. Additionally, it minimizes the occurrence of heat-induced distortions and defects that are more common in methods involving high-temperature processes. A unique feature of BJ is its capability to produce parts in multiple colors by utilizing different colored binders, enhancing the aesthetic appeal and functionality of printed objects for applications such as prototypes and functional parts. Moreover, the versatility of BJ in printing with a diverse array of materials—from metals to ceramics and polymers—broadens its applicability across various industries [45].

However, the BJ technique is not without its limitations. One of the main challenges is achieving high-density parts, as the process tends to produce objects with lower density due to the use of larger particle sizes in the powder [46]. The post-processing stage of BJ is also more labor-intensive and complex, involving multiple steps that may include curing, infiltration, and sintering to enhance the mechanical properties and density of the printed objects. Additionally, printed parts are susceptible to deformation during the removal of supports or during post-processing. Manual intervention is often required to manage and finish the prints, adding to the labor cost and time [47]. Another concern is the residue left by the binders, which can affect the surface finish and may require further post-treatment to remove. Lastly, the initial mechanical properties of the green bodies—the objects in their nascent stage post-printing—tend to be poor, necessitating additional processes to achieve the desired strength and durability [48].

#### 3.4.3. Materials of BJ

The BJ technology exhibits versatility in working with a wide array of materials, including polymers, metals, and ceramics, underscoring its adaptability across various manufacturing domains [49]. The selection of binder agents, predominantly composed of organic polymer-based materials, is critical and is primarily influenced by the binder solution’s wettability and bindability characteristics. This selection process is vital, as each binding agent possesses unique properties tailored to the specific type of powder it aims to bind. Commonly used polymers in this context include polyvinyl pyrrolidone (PVP), polyvinyl alcohol (PVA), and polyacrylic acid (PAA), each offering distinct advantages and considerations depending on the application and material compatibility [48].

#### 3.4.4. Applications of BJ

The diverse material compatibility of BJ technology enables its application across a multitude of sectors, significantly broadening its utility beyond conventional manufacturing paradigms. In the pharmaceutical domain, BJ technology is revolutionizing the production of oral medications, offering customized dosages and release profiles, which could lead to more personalized medicine practices [50]. Additionally, in dental medicine, the precision and adaptability of BJ facilitate the creation of ceramic dental prostheses, promising significant advancements in dental restoration and cosmetic dentistry [51]. Beyond medical applications, BJ’s capacity to work with metals, including 316L stainless steel, opens new avenues in manufacturing complex, high-strength components with applications ranging from aerospace to automotive industries, underscoring the technology’s versatility and potential to innovate across various manufacturing landscapes [52]. This breadth of application showcases the transformative potential of BJ technology in leading the next wave of manufacturing innovation, pushing the boundaries of what is possible in both medical and industrial applications.

#### 3.4.5. Developing Trends of BJ

Currently, the main goal with binder jetting is to find a way to eliminate the porosity that comes with the prints to make it a more viable option. One of the more recent developments and applications would be printing an oral medicine using binder jetting. For example, BJ is being used to print Spritam, which is a medicine used to treat seizures [50]. This advancement is very important because to be able to mass produce medicines of all types makes medicines more available to more people and lowers the cost of those medications, making it very relevant in today’s world.

### 3.5. Selective Laser Sintering

Selective laser sintering (SLS) technology stands out in 3D printing due to its ability to utilize a variety of materials and produce complex, functional parts with high accuracy. Unlike traditional 3D printing methods that rely on layers of material being deposited or cured, SLS employs a laser to selectively fuse powdered materials, eliminating the need for support structures and allowing for greater design freedom. Its versatility in material use, lack of need for support structures, and capability to create intricate, fully functional parts set SLS apart from other conventional 3D printing technologies.

#### 3.5.1. Principle of Operation of SLS

SLS technology utilizes lasers as the heat source and is a form of powder bed fusion. This technology is an effective method for rapid prototyping and forms a more solid layer of material with the high-density laser onto the bed of powder. This powder can be a variety of different materials and will be discussed further in the materials portion of this report. By printing parts one layer at a time and then building upon the previous layers, it is capable of rapidly producing 3D physical parts directly from a user-designed 3D model [53]. There are two classifications of SLS, including direct and indirect. These are based on the mechanism present in the specific SLS printer used [54]. A standard design for an SLS printer can be seen in a schematic form on Figure 8 [55].

The process follows these steps: the powder is filled, and the feed container is connected to the build cylinder, next a protective gas is passed into the forming room to reduce oxygen content, then the roller moves a thin layer of the powder over the build plate, and finally the laser will scan the powder surface and form the appropriate layers [53].

#### 3.5.2. Advantages and Limitations of SLS

The thermal and fluid behaviors of the layers during typical AM processes are essential to the proper adhesion between layers for the final product to be a strong and solid part. For SLS, this is a very good advantage, since the layer binding is very precise because of the use of a laser and results in the final product being almost isotropic [56]. This also allows SLS to be used for very complex component designs while having superior production time when compared with traditional manufacturing processes for the same designs, some of which would be unable to be used without AM. However, some disadvantages come from using powdered materials, which include porosity, shrinkage, impurities, and poor surface quality. Most parts will require post-processing for a clean final product [54].

#### 3.5.3. Materials of SLS

SLS is readily capable of producing 3D components utilizing a few materials, including plastics, composites, and ceramics [57]. Mainly, SLS is used for precise polymer production and is common in many industries for precise and rapid prototyping of more expensive components. The most commonly used polymer in SLS is nylon. In recent years, improvements to the post-processing of ceramic SLS components have made a viable final product. These improve some of the flaws originally found with the low density and strength of the original ceramic productions [58].

#### 3.5.4. Applications of SLS

With the rapid manufacturing time that SLS provides, it is a resourceful way to both prototype and produce fully functional polymer and ceramic parts that can be used in a variety of industrial applications. The first of which in this discussion involves the use of SLS in health fields. By using pharmaceutical powders, researchers have been able to create orally consumed medicines with different release times and could be an alternative in production in the short term [59]. The printing of pills is a very promising technique for the development of more patient-tailored medications to get the proper dosage rather than the standard sizes that currently exist [60]. SLS is also a beneficial technology with a variety of applications in aerospace as well. Mainly for the manufacturing of noncritical components as well as rapid prototyping of nonfunctional components. There are also instances where SLS is used as a functionally graded material due to the ability to produce SLS components with different crystallinities for different applications [61].

#### 3.5.5. Developing Trends of SLS

There have been recent developments in the creation of flexible electronics using SLS. The precise nature of the laser in the device allows it to be used in combination with metal nanoparticles for the creation of micropatterns, which can be used in small electrical devices as they take up minimal space [62].

In the SLS process, defects arise from the creation of gas bubbles due to overheating and trapped surrounding gasses. Artificial intelligence (AI) models are currently being developed to control the SLS device and decrease the amount of imperfections in the final product by performing different analyses and performing corrective responses to a condition in which a defect would result from [63].

### 3.6. Selective Laser Melting

Selective laser melting (SLM), also known as laser powder bed fusion (LPBF), is a type of powder bed fusion (PBF) printer that melts and fuses material powder together in layers [64]. Figure 9 represents the general components in a schematic found in all types of PBF printers. A heat source, either a laser or an electron beam, is used to fuse the first layer, that is the first cross-section of the part, covering the build platform. Then, the printing bed lowers by a layer for new powder to be spread over the previous using a roller or blade, which is often vibrated to promote a more even powder distribution, and the cycle repeats until the part is completed [65].

#### 3.6.1. Principle of Operation of SLM

SLM follows the same process as PBF from beginning to end in layer-by-layer construction, but only takes metals, typically pure, and specifically uses a high powered laser to completely fuse the powder to print the part. The SLM process begins with a fresh layer of metallic powder deposited onto the print bed using a roller or blade provided by the powder reservoir and any excess is caught by the overflow container. The building platform or print bed is constrained to a translation in the *z*-axis. The laser then activates. The beam is controlled by a scanner system, constrained to *x*–*y* rotation, which heats the powder bed to slightly above melting point to create a melt pool and draw the desired shape of the slicing plane. The molten metal then cools rapidly and solidifies, resulting in fused tracks. The stage starting when the laser interacts with the powder until the end of the molten metal’s solidification is called the Laser Powder Melt Pool (LPMP). After scanning the cross-section, the building platform lowers by an amount equal to the next layer’s thickness and the process repeats until the printed part is complete. After having fully cooled, the building platform is then raised to reveal the print and reduce lead time by freeing it from most of the powder [66]. During post-processing, the remaining unfused powder is retrieved and can be reused again and again [67]. Support structures of the recovered part are removed, and the part is cleaned with isopropyl alcohol or processed with CNC work to improve surface finish [68].

SLM parameters are controlled to minimize build time while conserving product quality. Scanning speed, hatch speed, layer thickness, and laser power are some of the most important to SLM [69]. The melting temperature the laser induces in the powder is controlled by laser parameters such as incident energy density (J/cm^2^) and laser power. Powder characteristics, including morphology, size, and apparent density, are important considerations in micro-structure parts. The typical layer thickness of commercial SLM systems is 20–100 μm, with particle size ranging from 20 to 50 μm [70]. The typical tolerance is ±0.2% with a lower limit of ±0.1 (±0.003″). The best materials for SLM are single-component metals including titanium, aluminum, and steel [71]. Melting metal can require temperatures as high as 1600 °C, which creates destabilization in the microscopic level. Therefore, support structures are needed due to the high residual stress and to mitigate the distortion. SLM is also widely used with cobalt-chromium, titanium alloys, nickel alloys, iron-based alloys, aluminum alloys, niobium alloys, refractory alloys, amorphous alloys, and super alloys [72]. There is a reason why SLM favors homogenous, elemental metals. While SLM has been used with alloys such as cobalt chromium, it is more feasible with monometallic powders because of the singular melting point [73]. Complete melting results in full fusion of the particles but sintering, in comparison, does not result in fully fused parts since the material remains slightly under the melting point [74]. Therefore, the mechanical properties of an SLM print are better because of increased particle merging and fuller density. The surface finish of SLM is also better [75]. Overall, SLM works well with pure metals and certain alloys, but will struggle to combine and process a heterogenous metal powder mixture.

#### 3.6.2. Advantages and Limitations of SLM

SLM, similar to other AM technologies, benefits from a shorter build time than traditional machining process. This is especially powerful when coupled with the fact that the process results in a fully functional object because of its high forming accuracy, net-shape ability, and high tolerancing as well as its ability to fabricate complex shapes [76]. The advantages of SLM are highly attributed to its ability to melt and fully fuse powder that results in fully dense parts—up to 100% density which allows it to maximize tensile strength (unlike sintering) to its full potential and allowing it to create light, but strong parts using metals such as aluminum. SLM is also suitable for visual models and prototypes. There is no need for binders and fluxing agents in the metallic powder [77]. The SLM process is often faster than selective laser sintering (SLS). One boon of SLM is the recyclability of its unmelted powder, with up to 99% powder recovery, that can virtually always be reused. SLA can produce functional, complex, net-shaped, and fully dense parts, but still struggles with limitations. The extreme heat is, again, a culprit of many problems seen with the nature of having to fully melt metal such as warping. It is a high-energy process that relies on temperature gradients to fuse particles [68], which requires higher energy costs and has been found to be less energy efficient than SLS by upwards of 10–20%. One paper on the quality control of SLM gives an in-depth review of the major factors that contribute to the formation of internal defects in SLM printed objects during the LPMP stage. SLM internal defects, with many related to higher temperatures, include impurities, lack of fusion, gas pores and micro-cracks, which are promoted by the balling, spatter, and keyhole phenomena. Defect formation can be suppressed by adjusting process parameters to the knowledge of the formation mechanism of each problematic phenomenon. SLM’s material flexibility, as addressed, is minimal because of its bias for single-component metals. While powder can act as an integrated support material in PBF, this does not apply to SLM because of its need for support structures due to the problems that arise from high temperatures. Furthermore, the build chamber must be filled with inert gas [78]. There is also a size limitation to SLA, with a max part size of about 280 × 280 × 325 mm^3^.

#### 3.6.3. Materials of SLM

The main materials utilized in SLM include titanium alloys (notably Ti6Al4V) for their exceptional strength-to-weight ratio and biocompatibility, making them ideal for aerospace and medical applications. Stainless steels (such as 316L and 304) are also widely used due to their corrosion resistance and mechanical properties, suitable for a wide range of industrial applications. Aluminum alloys, such as AlSi10Mg, offer advantages in automotive and aerospace parts because of their lightweight and good thermal properties. Cobalt–chrome alloys are chosen for high wear resistance in medical implants and aerospace components. Nickel-based superalloys, such as Inconel 625 and 718, are selected for their excellent strength and thermal resistance, crucial for applications in harsh environments. Finally, tool steels (e.g., H13, D2) are used for creating durable tooling components. Despite the broad utility of these materials, challenges such as controlling residual stresses, porosity, thermal crack, and ensuring material properties consistency remain key research and development focuses within the field.

#### 3.6.4. Applications of SLM

SLM sees application in the jewelry, dental, aerospace, automobile, and medical industries [79]. Its demand in high-tech areas and industrial interest stems from its ability to manufacture fully functional parts with excellent mechanical properties and high geometric complexity. Recently, SLM has now been able to open its doors to ceramics, such as alumina and zirconia, and to gradient materials. In 2010, Hagedorn was able to develop technology that completely laser melts pure ceramic powder and manufactures net-shaped specimens with almost 100% densities without requiring post-processing to achieve it [76]. AM technologies have been used by surgeons and material scientists to create patient-specific medical devices of any geometry and sometimes within a day [80]. Furthermore, while an even more recent paper demonstrates the success of an SLA created hip implant, the reliability of these biomedical parts needs to be investigated, especially the powder materials themselves, the corrosion behavior, and fatigue properties [81].

#### 3.6.5. Development Trends of SLM

SLM technology has seen significant advancements and developing trends aimed at overcoming its initial limitations and broadening its application spectrum. One of the foremost trends is the exploration and development of new material systems, including high-entropy alloys and functionally graded materials, to exploit unique property combinations for tailored applications [82]. Concurrently, there is a focus on optimizing process parameters through machine learning and AI to enhance part quality, reduce defects, and improve material properties predictively. Multi-laser systems have emerged to increase build rates and improve the efficiency of the SLM process, addressing productivity concerns for industrial-scale production. The integration of in situ monitoring and control systems using sensors and real-time data analysis aims to achieve consistent and reliable part quality by detecting and correcting process anomalies as they occur. Lastly, efforts towards achieving sustainability in SLM involve recycling of powder materials and energy efficiency improvements, crucial for minimizing the environmental impact of manufacturing processes. These trends underscore the dynamic evolution of SLM, pushing towards broader industrial adoption and the creation of more complex, high-performance components across sectors.

### 3.7. Direct Metal Laser Sintering

Direct metal laser sintering (DMLS) is another PBF process that is similar to SLM but only takes metal alloys or combined powder metals. DMLS was evolved from SLS and uses a laser beam to condense metal powder without binders, creating a high-density product. It can produce sophisticated components with complex geometries that casting, machining, and forming cannot because to their limitations [83].

#### 3.7.1. Principle of Operation of DMLS

DMLS feeds and deposits metallic powders using a core concept. It uses lasers to overlay metallic powders to make functioning parts. This method has three phases including powder delivery, deposition, and waste storage. In the powder delivery stage, a feeder with a roller and rake precisely applies material with the proper thickness to the preceding layer, as shown in Figure 10 during deposition. The deposition stage fuses metallic powder with the underlying layer by precisely moving the laser source [84]. Overhanging structures in fused layers are supported by powder not fused by high-energy sources [85]. In the waste storage stage, powder not used in deposition is stored for reuse. Looping through these phases creates a 3D component. The sintering mechanism uses heat to fuse powder particles into a dense component [86].

#### 3.7.2. Advantages and Limitations of DMLS

AM through DMLS stands out as a support-free technique that enhances its capacity to craft intricate pieces with precise dimensions [88]. This process facilitates the direct printing of metal components or prototypes using a diverse range of metal alloys while upholding their inherent qualities [89]. Moreover, the innovative feature of recycling unused metal powder post-printing elevates productivity in creating prototypes and components, concurrently curbing material costs and waste [90].

However, the utilization of DMLS in 3D projects entails considerable expenses, constituting one of the pricier avenues within AM due to machine and material costs [91]. Additionally, the nature of DMLS tends to produce components with higher porosity compared to traditional metal AM methods, albeit engineers have a degree of control over porosity during the printing process. Furthermore, when considering 3D printing projects constrained by build volume, engineers opt for the most suitable technology, recognizing DMLS as a choice for low-volume metal additive applications [92].

#### 3.7.3. Materials of DMLS

DMLS technology has applications in the construction of components utilizing a range of materials [93]. The materials that can be used in this technique are ABS nylon filled with glass and polymers of polycarbonate. In addition, DMLS is capable of working with a variety of metallic materials, such as copper, low-carbon steels, superalloys, and stainless steels. DMLS makes it easier to manufacture additive metal parts by providing compatibility with a wide variety of metals, including aluminum, titanium, steel, stainless steel, cobalt chrome, nickel alloys, and precious metals.

#### 3.7.4. Applications of DMLS

The manufacture of high-performance components is one of the primary applications for DMLS, which has seen widespread adoption in the aerospace and automotive industries. Additionally, it has applications in a wide variety of other industries, such as functional prototypes, tooling, and medical prostheses, to name a few. In addition, the use of DMLS in the process of manufacturing orthopedic implants for dogs is quickly becoming more commonplace. This broadens its uses beyond the realm of traditional implants to include individualized prostheses and bone replacements in veterinary medicine as well as other fields [94].

#### 3.7.5. Developing Trends of DMLS

In order to overcome the surface accuracy issues, a technique of optimization that is based on two criteria is presented. This strategy places an emphasis on optimizing certain subprocesses, such as the orientation of the component, the determination of the layer thickness, and the directions of the laser scanning, with the twin goals of achieving: (a) the shortest possible production time and (b) the fewest possible surface flaws. In addition to this, when it comes to the optimization process, the model takes into consideration the impacts that are caused by the shrinking of the material [95].

### 3.8. Electron Beam Melting

Electron beam melting (EBM) is an AM technique that uses a high-energy electron beam to fuse metal powder particles layer by layer to build complex parts.

#### 3.8.1. Principle of Operation of EBM

The process occurs inside a vacuum chamber to prevent oxidation of the materials and to maintain the integrity of the electron beam. A high-voltage electron beam is generated and focused onto a thin layer of metal powder, selectively melting the powder according to the digital design of the part. After one layer is melted and solidified, the build platform is lowered, and a new layer of powder is applied. This process repeats until the part is fully constructed [96]. The energy source, an electron beam, allows for rapid melting and solidification, making EBM distinct in its speed and the quality of the parts produced.

#### 3.8.2. Advantages and Limitations of EBM

EBM technology offers several benefits, including the ability to produce parts with complex geometries that are difficult or impossible to achieve with traditional manufacturing methods. It provides excellent material properties, comparable to wrought metals, due to the rapid cooling rates and high-temperature processing, which results in fine microstructures. EBM can process reactive metals such as titanium and its alloys in a vacuum, which is essential for aerospace and medical applications. Moreover, the process is relatively fast and efficient, with minimal material waste.

EBM also has its drawbacks. The requirement for a vacuum environment and the use of electron beam technology contribute to high equipment costs. The surface finish of EBM-produced parts is often rougher compared to other AM technologies, which may require additional post-processing. The selection of materials compatible with EBM is currently more limited than for other AM methods. Additionally, the layer-by-layer construction can introduce residual stresses, requiring heat treatment or other stress-relief processes.

#### 3.8.3. Materials of EBM

EBM technology primarily processes metals, with a focus on high-value, high-performance materials. Titanium alloys, such as Ti6Al4V, are among the most commonly used materials due to their strength, lightweight, and biocompatibility, making them ideal for aerospace and medical implants [97]. Nickel-based superalloys, such as Inconel 718, are used for applications requiring high strength and corrosion resistance at elevated temperatures. Cobalt–chrome alloys are also processed for their wear resistance and biocompatibility in medical prostheses. The general requirement for EBM precursor powder is stringent, emphasizing the need for high purity and uniform particle size to ensure consistent melting and solidification properties. This is crucial in metallic 3D printing, where the choice and quality of the powder significantly affect the final product’s mechanical properties and surface finish. Recent research is expanding the range of materials suitable for EBM, including refractory metals and high-entropy alloys, to broaden its application base [98].

#### 3.8.4. Applications of EBM

EBM has found applications in industries where the properties of metal parts are critical, and complexity is valued. In aerospace, EBM is used to manufacture lightweight structural components and complex engine parts that benefit from the weight reduction without compromising strength. The medical field uses EBM to create custom implants and prosthetic devices, taking advantage of the ability to produce porous structures that encourage bone ingrowth. The energy sector benefits from the production of durable parts for high-temperature applications. As the technology matures, its applications continue to expand into areas such as automotive and tooling, where customization and material properties are key considerations.

#### 3.8.5. Developing Trends of EBM

The future of EBM technology is focused on overcoming its current limitations while expanding its capabilities and applications. Research is underway to improve surface finish and reduce residual stresses, potentially broadening the range of materials that can be effectively processed. Advances in machine learning and artificial intelligence are being applied to optimize process parameters and predict outcomes, enhancing efficiency and part quality. There is also a trend towards multi-material processing, allowing for parts with graded properties or complex material systems. Sustainability efforts are focusing on reducing energy consumption and increasing material utilization rates. As the technology progresses, EBM is poised to play a significant role in the future of AM, with ongoing developments aimed at increasing its accessibility, versatility, and efficiency.

### 3.9. Multi-Jet Fusion

Multi-jet fusion (MJF) is a very innovative AM method created by the company Hewlett-Packard [99]. This type of AM method falls into the powder bed fusion family of printers alongside technologies such as SLS, DMLS/SLM, and EBM [100].

#### 3.9.1. Principle of Operation of MJF

MJF printers’ function in ways similar to most other 3D printing technologies but also has its own unique functions. MJF uses an array of inkjets to lay a layer of detailing agents one layer at a time to be fused in a bed of powdered material. These layers of detailing agents fuse together by use of a heating element [101]. This heating element functions by using an infrared lab that fuses materials in the same vein as the feed material. These various materials are thermoplastic polymers and are commonly used in such applications [102]. The step-by-step process of how a MJF printer operates is shown in Figure 11 [103]. Here, a material coating is applied that acts as the base layer for printing. Next, an agent is applied to the desired places in the print. Energy is then applied using the infrared heating element mentioned above, then the final fusion takes place, and the print is ready.

#### 3.9.2. Advantages and Limitations of MJF

When it comes to the MJF technology, it offers relatively low machining time with similar part properties with minimal post processing required after the fact [101]. Some studies, such as the AAPM study below, have reported MJF to be up to about 10 times faster than standard FDM printing for mass production of various parts [104]. When it comes to how the materials perform, MJF-printed designs often have a more ductile nature to them, which could be a desired material property for some applications but also a limitation in other applications depending on needs [105]. An MDPI study even found MJF printing to have an average printing cost of about half of other similar products. A wonderful photo can be seen below in Figure 12 of the various colors that MJF can print as well as the types of projects that can be printed out of MJF [106]. One of the MJF’s main limitations is its material limitations, because the ductility of MJF parts, while beneficial in some cases, may not be suitable for applications requiring higher material stiffness or strength. Besides, the advantages of MJF, such as its speed and cost-effectiveness, are maximized in particular scenarios, such as mass production, which may not apply to all projects.

#### 3.9.3. Materials of MJF

MJF technology employs an extensive selection of powdered materials as the foundation for its printing process. These materials are meticulously fused by the application of two distinct types of binder fluids: a fusing agent and a detailing agent. The detailing agent, often incorporating a cyan, magenta, yellow, and key (CMYK) color scheme, plays a crucial role in enabling the production of parts in a vast spectrum of colors, adding to the versatility and aesthetic appeal of the printed objects. This capability was exemplified in Figure 12, showcasing the technology’s ability to achieve detailed and vibrant colorations in printed parts. The fusion of these agents with the base materials under precise conditions not only ensures the structural integrity of the components but also opens up new possibilities for customization and design innovation in AM [107].

#### 3.9.4. Applications of MJF

The potential applications of MJF go far and wide, but in part to being a bit more energy and resource efficient than SLS, MJF can be more attractive to industries that require more energy and resource efficiency [108]. MJF printed parts are said to have significant vibration isolation characteristics, especially between the ranges of 10–30 Hz, which means that MJF parts could be desired for various applications that require that specific kind of isolation within that range [109].

#### 3.9.5. Developing Trends of MJF

Some common developing trends in the world of MJD 3D manufacturing would be to use MJF parts in mechanical parts/devices along with biometrical lattices, structures, and other medical and orthotics applications such as prosthetics. The MJF technology was also found to have a potential in creating mechanical tools and well as in devices that are required to be fluid-tight [110]. These applications are only scratching the surface as to what this technology has to offer. The future of MJF is bright and shows potential to take a significant part of certain industries in the future with its abilities to print efficiently as well as print parts with certain desirable material properties. MJF, as most 3D printing technologies, is still more or less in its infant stage and could see some major growth in the coming years.

### 3.10. Direct Energy Deposition

Direct energy deposition (DED) is an advanced AM technology that involves the use of focused thermal energy—such as a laser, electron beam, or plasma arc—to fuse materials by melting as they are being deposited. DED is distinguished by its ability to create high-quality metal parts directly from a digital model, offering significant advantages in terms of material efficiency, design flexibility, and the ability to repair or add material to existing components. This report explores the principle of operation, advantages and limitations, materials, applications, and developing trends of DED technology.

#### 3.10.1. Principle of Operation of DED

DED technology operates by focusing thermal energy to melt a material—typically metal powder or wire—as it is being deposited onto a substrate or part surface. The material is delivered through a nozzle that moves in multiple axes, allowing for the construction of complex geometries. Figure 13 shows the principle of operation of two different types of DED technologies [111].

The process is typically controlled by a CAD file, which guides the nozzle’s path to create the desired shape layer by layer. The precise application of energy and material results in parts that are fully dense and strongly bonded to the substrate [112].

#### 3.10.2. Advantages and Limitations of DED

There are several advantages of DED. DED minimizes waste by depositing material only where needed. DED also enables the creation of complex geometries that are difficult or impossible to achieve with traditional manufacturing methods. DED can be used to add material to existing parts, allowing for the repair of worn or damaged components, or the modification of parts to improve performance. DED is capable of processing a wide range of materials, including difficult-to-process alloys [113]. For certain applications, DED can be faster than other AM technologies, especially when producing large or medium-sized components.

DED has some limitations, such as rough surface finish and low precision. DED often requires post-processing to achieve a smooth surface finish, as the parts may have a rough texture directly after manufacture. While suitable for many applications, DED may not achieve the fine detail or dimensional accuracy of some other AM processes.

#### 3.10.3. Materials of DED

DED technology boasts a remarkable capacity to handle an expansive range of materials, making it a versatile tool in the AM landscape. This technology is proficient in processing a wide spectrum of metals, such as stainless steel, titanium alloys, nickel-based alloys, aluminum, and cobalt–chrome, catering to the demands of various industrial applications. Beyond metals, DED also extends its capabilities to ceramics and composite materials, although these applications are less widespread [114]. For DED technology, the precursor powder must meet specific requirements in terms of particle size distribution, flowability, and purity to ensure efficient deposition and optimal material properties in the final product. The precision in powder characteristics is vital to achieving the desired material behavior during the melting and solidification processes, which is instrumental in extending DED’s application in manufacturing complex and custom fabrications across various industries. Recent advancements in DED technology are continually broadening the scope of its material compatibility, paving the way for its application in more specialized fields. These developments not only enhance the utility of DED, but also underscore its potential to revolutionize manufacturing processes by accommodating a diverse array of materials for custom and complex fabrications [115].

#### 3.10.4. Applications of DED

DED technology is widely applied in several key industries, demonstrating its versatility and effectiveness. In the aerospace sector, it is utilized for manufacturing and repairing vital components such as turbine blades and structural parts, underscoring its importance in maintaining the reliability and performance of aircraft. The energy industry benefits from DED through the repair of parts used in oil and gas exploration as well as in power generation equipment, highlighting its role in sustaining critical infrastructure. In the medical field, DED’s ability to produce customized implants and prosthetics showcases its potential in personalized medicine, offering solutions tailored to individual patient needs. The tooling industry uses DED to add features or coatings to tools, dies, and molds, enhancing their functionality and lifespan. Additionally, its flexibility in processing different materials makes DED invaluable for research and development purposes, particularly for prototyping and material development, thereby fueling innovation across various sectors.

#### 3.10.5. Developing Trends of DED

The evolving landscape of DED technology is marked by several promising developments that are set to enhance its capabilities and broaden its applications across industries. Hybrid manufacturing emerges as a notable trend, integrating DED with traditional subtractive manufacturing within a single setup to harness the combined strengths of both methods, optimizing the production process for efficiency and precision. Concurrently, advancements in DED are paving the way for multi-material printing, enabling the creation of components with multiple materials in a single build. This innovation introduces the potential for parts with functionally graded materials and localized property variations, expanding the design possibilities and functional capabilities of manufactured parts. Furthermore, the integration of real-time monitoring and adaptive control systems is revolutionizing the DED process, significantly improving part quality and process reliability through precise control and adjustments during fabrication. Complementing these technological strides, the development of sophisticated software and simulation tools is crucial, offering predictive insights and optimization strategies for material properties and process parameters, thereby minimizing the reliance on trial and error and enhancing the overall quality of the final products. Collectively, these developments signify a significant leap forward for DED technology, showcasing its growing importance in the AM domain and its potential to revolutionize manufacturing practices across a diverse array of sectors by offering unmatched material efficiency, design flexibility, and the capability to repair or augment existing components.

### 3.11. Carbon Fiber Reinforcement

Carbon fiber reinforcement (CFR) technology combines carbon fiber with polymer matrixes to produce composite materials that are exceptionally strong and lightweight.

#### 3.11.1. Principle of Operation of CFR

The principle of operation involves the selective layer-by-layer deposition of carbon fiber-infused filaments, which are then melted and fused together using a heated nozzle in a process similar to FDM. This technique allows for the precise alignment of carbon fibers, optimizing the strength-to-weight ratio of the final product [116]. Advanced variants of this technology can also embed continuous carbon fiber strands into a base material, significantly enhancing the mechanical properties of the printed object. Figure 14 shows 3D-printed bicycle lugs reinforced with continuous carbon fiber [117].

#### 3.11.2. Advantages and Limitations of CFR

CFR offers several compelling advantages, including the production of parts that are significantly stronger and lighter than those made from conventional materials, making it ideal for high-performance applications. The technology allows for complex geometries that are difficult to achieve with traditional composite manufacturing techniques, along with reduced waste and shorter production cycles [118]. Additionally, the ability to customize the fiber orientation provides unparalleled control over the mechanical properties of the final part.

Despite its benefits, CFR technology faces limitations such as higher material and operational costs compared to standard AM processes. The range of materials compatible with carbon fibers is also limited, primarily to certain thermoplastics and thermosetting polymers. Moreover, the technology requires specialized equipment and expertise to ensure the optimal placement of fibers and to manage the thermal stresses during printing, which can affect the dimensional accuracy and surface finish of the parts.

#### 3.11.3. Materials of CFR

In CFR, the choice of materials plays a pivotal role in achieving the desired mechanical and thermal properties of the final product. The base matrix typically involves high-performance thermoplastic polymers such as nylon, PEEK, and ABS. These polymers are chosen for their excellent balance of strength, durability, and ease of processing. The reinforcement comes in the form of short or continuous strands of carbon fiber, which are embedded into the polymer matrix to enhance the composite material’s overall properties [119]. The inclusion of carbon fiber, particularly in a continuous form, significantly elevates the mechanical performance of the composite. Continuous carbon fiber reinforcement leads to a marked increase in tensile strength and rigidity, which are critical for components subjected to high stress or load. Moreover, these carbon fiber-reinforced materials exhibit superior thermal stability, maintaining their integrity and performance over a wide temperature range. This combination of high strength, stiffness, and thermal resistance makes CFR composites ideal for use in advanced engineering applications, including aerospace, automotive, and high-performance sporting equipment, where materials must withstand rigorous operational conditions while minimizing weight and maximizing performance.

#### 3.11.4. Applications of CFR

CFR is increasingly employed in industries where strength-to-weight ratio is critical, such as aerospace, automotive, and sporting goods. In aerospace, it is used for producing lightweight structural components and fixtures. Automotive applications include the fabrication of high-performance parts such as gears, brackets, and structural components that benefit from the reduced weight without sacrificing strength. In the sporting goods industry, CFR technology is used to create equipment such as bicycles, racquets, and drones, offering enhanced performance characteristics.

#### 3.11.5. Developing Trends of CFR

The future of CFR is driven by ongoing innovations aimed at expanding its applications and improving performance. One key trend is the development of new composite materials that offer higher thermal and chemical resistance, opening up new industrial applications. Another area of focus is the improvement in printing technologies to enable the use of continuous carbon fiber reinforcement over larger areas and more complex geometries, further enhancing the mechanical properties of printed parts. Additionally, advancements in simulation and modeling software are improving the predictability and optimization of fiber orientations, reducing the need for physical prototypes and accelerating the design-to-production cycle. As these trends progress, CFR is set to revolutionize industries by providing lighter, stronger, and more customizable components.

### 3.12. Laminated Object Manufacturing

Laminated Object Manufacturing (LOM) involves the layer-by-layer bonding of sheet materials, which are precisely cut to shape and then bonded together to form a three-dimensional object.

#### 3.12.1. Principle of Operation of LOM

The process of LOM starts with a roll or a stack of sheets made of metal, paper, or plastic. Each sheet is affixed to the layer beneath using an adhesive, welding, or another bonding technique. A laser or knife then precisely cuts the outline and internal features of the current layer based on the digital model. This process repeats, stacking and shaping layers until the desired 3D object is fully formed. Excess material from each layer acts as support for the object during the build and can be removed afterward [120].

#### 3.12.2. Advantages and Limitations of LOM

LOM offers several advantages, including the ability to use a wide range of materials, from paper and polymers to metals, making it versatile across various applications. It is cost-effective, especially when using inexpensive materials such as paper, and the process can be faster than other AM techniques for certain geometries and materials. Additionally, the waste material can often be recycled, especially when paper is used, making it an environmentally friendly option. The technology is also beneficial for creating large parts and offers unique aesthetic finishes, particularly with paper-based laminates [121].

However, the technology faces limitations, including lower dimensional accuracy and part strength compared to other AM methods, due to the layer bonding mechanism. The range of materials suitable for LOM is broad but not as extensive in terms of composite or functionally graded materials. Furthermore, the post-processing required to remove excess material and achieve the final part geometry can be labor-intensive and time-consuming. The surface finish may also be rough, requiring additional post-processing to smooth out.

#### 3.12.3. Materials of LOM

LOM can process a diverse array of materials, including paper, metal foils, various plastics, ceramic-based materials [122], and composite materials [123]. Paper is commonly used for conceptual models and visual prototypes due to its low cost and ease of handling. Metal foils allow for the creation of more durable and functional prototypes or parts, though they require stronger bonding techniques such as ultrasonic welding. Plastics offer a balance between durability and ease of processing, suitable for both prototypes and functional parts. The choice of material significantly influences the application and performance of the finished product.

#### 3.12.4. Applications of LOM

The applications of LOM are varied and span several industries. In prototyping, especially for visual and conceptual models, paper-based LOM is popular due to its cost-effectiveness and rapid turnaround. Metal and plastic laminates are used in the automotive, aerospace, and electronics industries for creating functional prototypes, tooling, and even end-use parts that benefit from the layered aesthetic or specific material properties. Additionally, the education and architectural sectors utilize LOM for creating detailed scale models and teaching aids, capitalizing on its ability to produce large objects at relatively low costs.

#### 3.12.5. Developing Trends of LOM

The field of LOM is witnessing several developing trends aimed at overcoming its limitations and broadening its application scope. One significant trend is the integration of advanced materials, including composites and functionally graded materials, to enhance part strength and functionality. There is also a growing focus on improving bonding techniques to increase durability and dimensional accuracy. Innovations in cutting technology, such as more precise lasers and knives, are enabling finer details and better surface finishes. Additionally, the development of automated post-processing solutions is aiming to reduce the time and labor associated with removing excess material. As these trends evolve, LOM is expected to become more competitive with other AM technologies, offering unique advantages in cost, speed, and material diversity.

## 4. Developing Trends of Additive Manufacturing

AM is evolving rapidly, with key trends shaping its future. Figure 15 shows the developing trends of AM technologies. 3D bioprinting is advancing the fabrication of complex biological structures, promising revolutionary healthcare solutions. Three-dimensional food printing is transforming culinary arts, allowing for customized nutrition and intricate food designs, merging gastronomy with technology for personalized eating experiences. Meanwhile, large-scale 3D printing is pushing the boundaries of construction and industrial production, enabling the creation of large structures and components with unprecedented speed and efficiency. 4D printing introduces time as a dimension, enabling printed objects to change shape or function in response to stimuli, offering innovative applications in smart materials and structures. AI-based 3D printing is optimizing the manufacturing process, enhancing design, predicting material behavior, and improving the quality and efficiency of printed objects, marking a significant leap toward intelligent and autonomous production systems. There are also other directions, such as multi-material and new materials.

### 4.1. 3D Bioprinting

Biological additive manufacturing, also known as biological fabrication, integrates principles of biology, chemistry, and engineering to fabricate biological structures and materials via AM techniques [128]. This innovative field leverages the precision and versatility of 3D printing technologies to create complex biological constructs, such as tissues, organs, and biocompatible materials, layer by layer [129,130]. State-of-the-art advancements include the development of sophisticated bioprinting methods capable of handling living cells, bioinks composed of natural or synthetic biomaterials that support cell growth and differentiation, and the creation of vascularized tissues that mimic natural blood vessels for improved survival and function of fabricated tissues. Challenges remain, including the replication of the complex microarchitecture of native tissues, ensuring long-term viability and integration of printed constructs within the body, and scaling up the technology for widespread clinical application [131].

Significant efforts are underway to overcome these hurdles, with ongoing research focusing on improving the resolution and fidelity of bioprinted structures, developing new bioink formulations, and enhancing the mechanical and functional properties of printed tissues. Figure 15a shows a 3D-bioprinted human tissue (an ear) with biodegradable plastic scaffolding using an integrated tissue and organ printing system (ITOP) by the Wake Forest Institute for Regenerative Medicine [124]. Besides, bioinspired AM is a cutting-edge convergence of biology, material science, and advanced manufacturing techniques. This innovative approach draws inspiration from the natural world, aiming to mimic the complex structures and functionalities observed in biological systems through AM processes [132].

### 4.2. 3D Food Printing

Three-dimensional food printing represents a pioneering intersection of technology and gastronomy, offering personalized nutrition, intricate edible designs, and the potential for sustainable food production. State-of-the-art trends in this domain involve the use of a variety of edible materials, including proteins, carbohydrates, and fats, which are extruded layer by layer to create complex and customized food items. This technology not only allows for the customization of nutritional content to meet individual dietary needs, but also introduces novel textures and forms in culinary arts, potentially reducing food waste by utilizing ingredients more efficiently. Despite its innovative prospects, 3D food printing faces significant challenges, such as the limited range of printable food materials, slow printing speeds compared to conventional cooking methods, and the need for stringent food safety standards. Additionally, the acceptance of 3D-printed foods by consumers and the integration of such technologies into commercial kitchens pose further hurdles. Ongoing research and development aim to overcome these obstacles by enhancing printer technology, expanding the variety of printable food materials, and improving the scalability of 3D food printing processes [125]. Figure 15b shows a 3D-printed pizza by Beehex through a fund from NASA [125].

Optimizing the rheological properties of 3D printing food products presents unique challenges, as these properties are crucial for both the printability of food pastes and the quality of the final product. Rheological properties, including viscosity, yield stress, and thixotropy, must be carefully balanced to ensure that food materials can be extruded smoothly through the printing nozzle and maintain their shape once deposited. Some challenges in optimizing these properties are discussed as follows.

Complexity of Food Materials: Food materials are inherently complex and can exhibit non-Newtonian behavior, making it challenging to predict how they will behave under the shear rates applied during 3D printing. Ingredients can interact in unpredictable ways, affecting their rheological properties.Variability of Ingredients: Natural variability in food ingredients due to factors such as origin, season, and processing can lead to fluctuations in the rheological properties of the food paste, affecting consistency and printability.Sensitivity to Processing Conditions: The rheological properties of food materials can be highly sensitive to temperature, pH, and the presence of air bubbles, among other factors. Maintaining optimal conditions throughout the printing process to ensure consistent properties is challenging.Balancing Printability with Nutritional and Sensory Qualities: Achieving the right rheological properties for printing often requires the addition of modifiers such as hydrocolloids or emulsifiers. However, these adjustments can impact the nutritional value, taste, and texture of the final product, potentially compromising consumer acceptance.Post-Processing Changes: Foods can undergo changes in structure and rheology after printing or during cooking or cooling processes. Predicting and controlling these changes to ensure the final product meets desired quality standards is difficult.Equipment Limitations: The performance and characteristics of 3D food printers, especially the extrusion system, can limit the range of rheological properties that can be effectively printed. Printers with enhanced capabilities can be expensive and less accessible.

Addressing these limitations requires a multidisciplinary approach, combining food science, material science, and engineering. Advances in ingredient technology, precise control systems for 3D printers, and a deeper understanding of the rheology of complex food systems are essential for overcoming these challenges and unlocking the full potential of 3D food printing.

### 4.3. Large-Scale 3D Printing

Large-scale additive manufacturing has emerged as a groundbreaking technology, significantly impacting the construction, aerospace, and manufacturing sectors by enabling the direct fabrication of large structures, components, and parts. State-of-the-art advancements include the development of large-format 3D printers capable of printing entire buildings or large aerospace components from specialized materials, ranging from concrete and metals to composite polymers. This technology offers the potential for increased customization, reduced waste, and accelerated construction times [133]. However, challenges persist, including the need for improved mechanical properties of printed materials, the scaling up of printing processes without compromising precision or material properties, and the integration of large-scale 3D printing into existing manufacturing and construction ecosystems. Additionally, there are logistical challenges in transporting and operating large-scale printers on-site and environmental concerns related to the energy consumption and sustainability of the materials used [134]. Ongoing research and development aim to address these issues by enhancing material formulations, printing technologies, and process efficiencies, paving the way for broader adoption and innovation in large-scale AM. Figure 15c shows the Crane Wasp 3D printer, created to print homes using local materials [126].

Integrating structural reinforcement into large-scale 3D printed components presents significant challenges that necessitate meticulous attention to the material and process intricacies. The mechanical strength and durability of printed structures, especially in sectors demanding high reliability such as aerospace and construction, require advancements in reinforcement techniques and materials. Moreover, the absence of standardized guidelines and regulations specifically tailored to large-scale AM poses a hurdle to ensuring uniform quality, safety, and performance across the industry. Establishing comprehensive standards and regulatory frameworks is essential for fostering innovation, ensuring structural integrity, and promoting widespread acceptance and integration of large-scale 3D printing technologies into traditional manufacturing and construction practices.

### 4.4. 4D Printing

Four-dimensional printing, an innovation extending beyond traditional 3D printing, incorporates the dimension of time to create objects that can change shape, properties, or functionality in response to external stimuli such as temperature, light, moisture, or magnetic fields. This dynamic technology leverages smart materials, including shape-memory polymers and responsive hydrogels, to fabricate objects with pre-programmed transformations [135]. Shape memory alloys (SMAs) represent a pioneering class of smart materials in 4D printing, offering distinct advantages due to their ability to return to a pre-defined shape when subjected to an appropriate thermal stimulus. These materials, including nickel–titanium alloys, can undergo phase transitions that enable them to remember and revert to their original shapes upon heating [136]. Their integration into 4D printing opens avenues for creating complex, self-adjusting structures and devices that can react to temperature changes with high precision. The exploration of SMAs in 4D printing highlights the potential for advanced applications in fields requiring adaptive materials, such as responsive aerospace components and medical devices that can conform to anatomical changes, further pushing the boundaries of what is achievable with AM technologies.

State-of-the-art developments in 4D printing have led to potential applications in various fields, including biomedical devices that adapt to bodily changes, self-assembling structures for aerospace and construction, and smart textiles that adjust to environmental conditions [137]. Despite its promise, 4D printing faces challenges, including the limited range of smart materials available, the need for precise control over material properties to ensure predictable and uniform responses, and the complexity of designing objects that can perform intended functions under real-world conditions. Ongoing research aims to address these challenges by developing new materials, refining AM techniques, and enhancing computational design tools to accurately predict and program the dynamic behavior of printed objects. Figure 15d shows a ’smart’ self-folding material that can transform shape, printed by MIT’s Self-Assembly Lab [127].

### 4.5. AI-Based Additive Manufacturing

AI-based additive manufacturing integrates advanced technologies such as AI, machine learning, and the Internet of Things (IoT) with traditional 3D printing processes to enhance efficiency, quality, and customization. This intelligent approach enables real-time monitoring and control of the printing process, predictive maintenance of equipment, and dynamic adjustment of printing parameters for optimal results [138]. State-of-the-art trends in AI-based AM involve the use of AI algorithms to predict material properties and outcomes, IoT for seamless integration of AM systems into broader production networks, and digital twin technologies for virtual simulation and optimization of the manufacturing process [139]. Challenges facing AI-based AM include ensuring data security and privacy, managing the complexity of integrating multiple technologies, and overcoming the resistance to adopting new manufacturing paradigms within established industries. Furthermore, there is a continuous need for skilled workers capable of operating these advanced systems and for ongoing development to reduce costs and improve the scalability of smart AM solutions. Research and development efforts are focused on addressing these challenges, aiming to unlock the full potential of AI-based AM across various sectors. AI and machine learning algorithms optimize printing parameters in real time, enhancing quality, reducing material waste, and improving the mechanical properties of printed parts. Simulation software allows for the predictive modeling of material behavior and the optimization of designs before printing, saving time and resources.

### 4.6. Innovations in Materials, Quality Control, and Post-Processing

The ability to print with multiple materials simultaneously or to combine different AM processes in a single machine (hybrid printing) is a significant area of development. This advancement enables the creation of parts with varied properties (e.g., rigid and flexible, conductive and insulative) within a single build process, expanding the functional capabilities of AM parts. Hybrid systems that integrate subtractive processes (e.g., CNC milling) with additive processes improve surface finishes and dimensional accuracy. The exploration and development of new materials specifically designed for AM processes are expanding the capabilities and applications of AM. This includes the creation of high-performance polymers, metals, ceramics, and composite materials. Material innovation also focuses on sustainability, with an increasing emphasis on recyclable and biobased materials.

Improvements in build size and printing speed are crucial for the broader adoption of AM across industries. Developments in this area include the creation of larger print beds, faster printing techniques, and more efficient post-processing methods. These improvements aim to make AM viable for mass production and large-scale applications, moving beyond prototyping and small-batch production. Enhancements in the precision and repeatability of AM processes are vital for meeting industrial quality standards. This involves the development of more sophisticated monitoring and control systems that can detect and correct errors in real time during the printing process. Such systems rely on sensors and machine vision technologies to ensure parts are built to specification, reducing failure rates and improving reliability.

Advancements in post-processing are crucial for advancing AM from a specialized to a mainstream production technique, especially important in sectors such as aerospace and healthcare. Automated support removal, using mechanical, chemical, or thermal methods, now allows for efficient and precise part cleaning. Surface finishing technologies, such as chemical vapor smoothing and electro-polishing, are reducing manual work and improving both the looks and function of 3D printed parts by minimizing surface roughness. Furthermore, techniques such as hot isostatic pressing (HIP) and annealing are being automated to enhance the mechanical properties of 3D printed parts, addressing common issues such as stress and porosity, thereby making AM more viable for a wide range of applications.

### 4.7. Standardization and Regulatory

Addressing standardization and regulatory challenges in AM is crucial for unlocking its full potential across various industries. The absence of universally accepted standards and clear regulatory guidelines currently presents significant obstacles to the widespread adoption of AM technologies. These challenges stem from the innovative nature of AM, which diverges significantly from traditional manufacturing processes in terms of materials, production techniques, and product properties. Standardization efforts are essential for ensuring consistency, safety, and quality in AM-produced goods. Without standardized protocols, comparing and validating the performance of products made through AM across different sectors becomes problematic. This lack of standardization also hampers material development, process optimization, and the sharing of knowledge within the AM community, ultimately slowing down innovation. On the regulatory front, establishing clear guidelines is necessary to address safety concerns and intellectual property issues, which are paramount for consumer and industry confidence. For instance, in highly regulated sectors such as aerospace, automotive, and medical devices, manufacturers must navigate complex certification processes that are not yet fully adapted to the specifics of AM technologies. This adaptation is essential for ensuring that AM products meet stringent safety and performance requirements.

Moreover, there is a pressing need for regulations that encompass the entire lifecycle of AM products, from design and production to use and end-of-life disposal. Such comprehensive regulatory frameworks could address environmental concerns, including energy consumption and material waste, promoting sustainability in AM practices. Progress in overcoming these standardization and regulatory hurdles is being made through the collaboration of industry stakeholders, research institutions, and regulatory bodies. These efforts include the development of industry-specific standards, the creation of certification pathways for AM materials and processes, and the initiation of dialogue on ethical considerations and intellectual property rights in the context of AM.

In conclusion, tackling the challenges of standardization and regulation is key to facilitating the broader adoption of AM technologies. By establishing clear, universally accepted standards and regulatory frameworks, the AM industry can ensure product safety and quality, foster innovation, and build the trust of consumers and industries alike, paving the way for a new era of manufacturing.

### 4.8. Circular Economy and Sustainability

The economic analysis of AM processes reveals a complex landscape of costs, benefits, production efficiencies, and return on investment (ROI), varying significantly across different AM technologies and application contexts. Understanding these financial implications is crucial for businesses considering adopting AM processes.

Initial costs for AM include the purchase of printers, materials, and software, alongside operational costs such as energy consumption, maintenance, and labor. Advanced AM systems, especially those capable of metal printing, can require substantial initial investment. Material costs can also be high, particularly for specialized or high-quality inputs. Training and integrating AM into existing production workflows can add to the initial expenses. AM offers several financial benefits, including reduced material waste, lower inventory and storage costs, and the ability to produce complex designs without a significant increase in cost. For customized or low-volume production, AM can significantly reduce the cost per unit compared to traditional manufacturing methods. Technology also enables faster product development cycles, reducing time to market and potentially increasing competitive advantage. AM can enhance production efficiency by consolidating parts, thereby reducing assembly time and costs. It allows for just-in-time production, minimizing inventory levels and associated costs. However, the production speed of AM may be slower compared to traditional manufacturing methods for high-volume production, potentially offsetting some efficiency gains unless combined with other manufacturing processes in a hybrid approach.

The return on investment (ROI) from different AM technologies can vary. FDM is often used for prototyping due to its lower cost, FDM can offer a quick ROI in environments where design flexibility and rapid prototyping are valued. SLS and DMLS are more expensive but can produce functional parts with properties close to traditionally manufactured items. They are more likely to be used in final product manufacturing, potentially offering a higher ROI by enabling the production of complex, high-value items. SLA and MJ are ideal for producing high-detail parts with smooth surface finishes, SLA can be cost-effective for applications requiring precise geometries and aesthetics, such as dental or medical models. The ROI also depends on the industry. In aerospace and medical sectors, where the cost of failure is high, and the value of customized, complex parts is significant, the ROI can be substantial. In contrast, for industries focused on high-volume, low-margin products, integrating AM may require a more strategic approach to realize a positive ROI.

The financial implications of adopting AM are influenced by a myriad of factors, including the type of AM technology, the scale of adoption, the specific application, and the industry. While the initial costs can be high, the benefits of customization, reduced lead times, and efficiency improvements present a compelling case for AM in many scenarios. A thorough cost–benefit analysis, considering both direct and indirect financial impacts, is essential for businesses to make informed decisions about investing in AM technologies. The strategic integration of AM can offer competitive advantages, ultimately leading to a positive return on investment.

The development of AM technologies is increasingly aligned with the principles of the circular economy. This includes the use of recycled materials in the printing process, the development of recyclable or biodegradable materials, and the potential for AM to reduce waste through optimized design and on-demand production. The direction of development in AM technologies and processes reflects a push towards greater integration with digital technologies, enhanced material capabilities, and a focus on sustainability. These trends aim to solidify AM’s role in the future of manufacturing, offering unparalleled flexibility, efficiency, and innovation across industries.

## 5. Conclusions

This study has provided an extensive overview of the AM landscape, reflecting on its historical evolution, current state, and future prospects. AM technologies, characterized by their versatility and innovation, offer significant advantages across various sectors, including aerospace, healthcare, automotive, and fashion. The exploration of different AM technologies reveals a nuanced understanding of their operational mechanisms, material requirements, and application potentials. Despite the transformative impacts of AM, challenges such as printing speed, material diversity, cost, and post-processing requirements persist, indicating areas for future improvement and research.

Future developments in AM are poised to address these challenges through advancements in material science, printing technologies, and process optimization. The integration of artificial intelligence and the exploration of new domains such as 3D bioprinting, 3D food printing, and large-scale 3D printing are expected to further enhance the efficiency, accessibility, and applicability of 3D printing technologies. Moreover, the potential for multi-material printing and improved precision offers a glimpse into a future where 3D printing could seamlessly blend into traditional manufacturing processes, offering customized and on-demand production capabilities that were previously unattainable. Standardization, regulation, circular economy, and sustainability also need future efforts and studies.

In conclusion, AM stands at the cusp of a new era in manufacturing, with the potential to fundamentally alter how products are designed, produced, and distributed. As the technology continues to evolve, overcoming its current limitations, its integration into the broader manufacturing ecosystem will likely accelerate, heralding a future where AM plays a central role in the next manufacturing revolution. The ongoing research and development within this field are critical, not only for advancing the technology itself but also for realizing the full spectrum of its applications across industries worldwide.

## Figures and Tables

**Figure 1 sensors-24-02668-f001:**
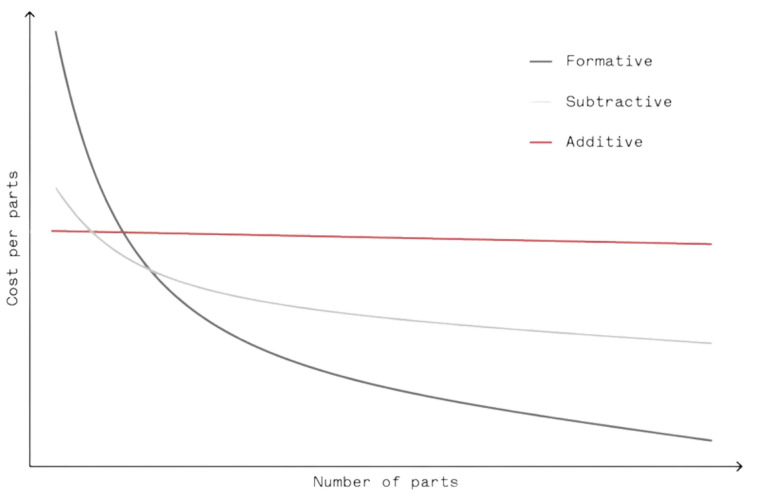
Cost comparison between formative, subtractive, and additive manufacturing [3].

**Figure 2 sensors-24-02668-f002:**
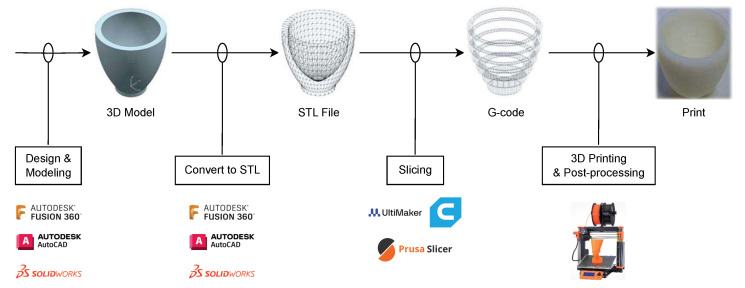
Overall workflow of the additive manufacturing process.

**Figure 3 sensors-24-02668-f003:**
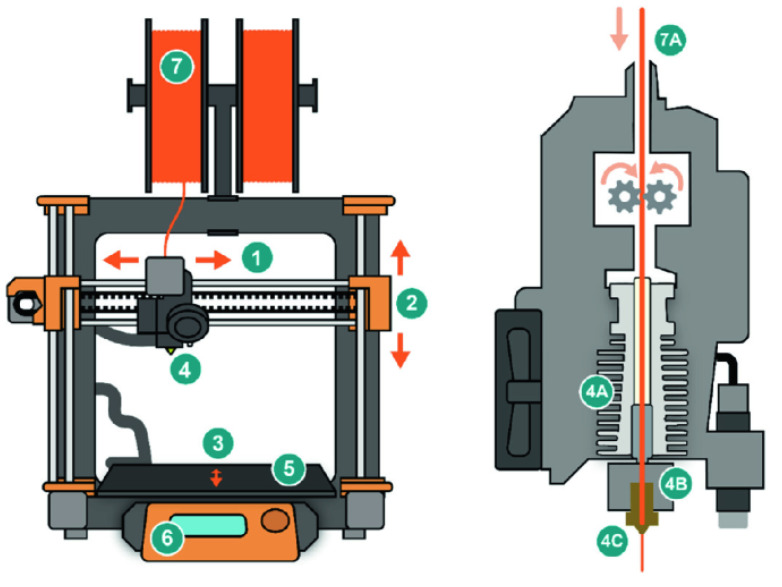
Schematic diagram of FDM 3D printing technology [15]. The picture on the left shows the whole structure of typical FDM printers where parts (1) to (7) are x-axis motor, z-axis motor, y-axis motor, hot nozzle, printing bad, controller display board, and filaments, respectively. The picture on the right shows more details of the printing nozzle where (7A) is the feed filament, and (4A), (4B), and (4C) are heating wires, Hotend, and extruded materials, respectively.

**Figure 4 sensors-24-02668-f004:**
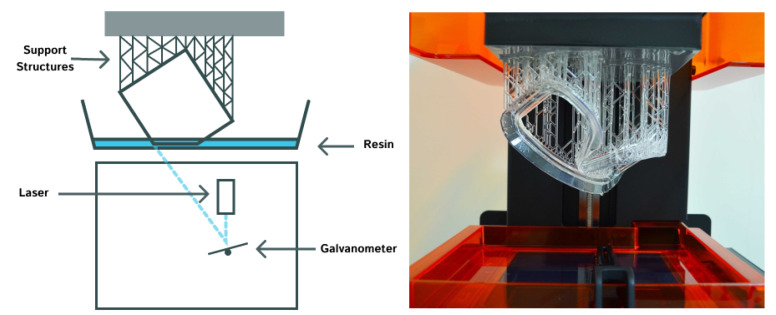
Schematic of the SLA 3D printing process and a print sample [29].

**Figure 5 sensors-24-02668-f005:**
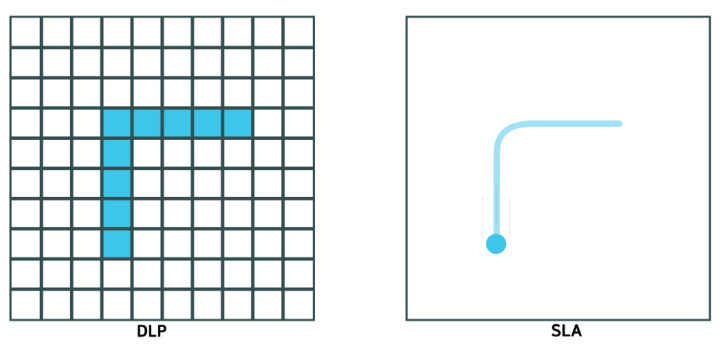
Different lighting technologies between SLA and DLP [29].

**Figure 6 sensors-24-02668-f006:**
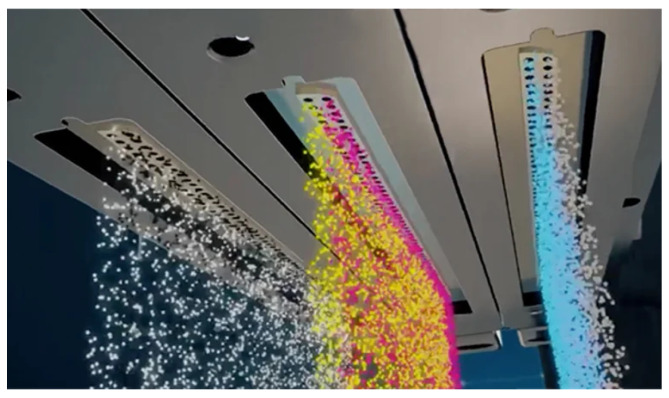
An illustration of how the Stratasys J55 dispenses print material [38].

**Figure 7 sensors-24-02668-f007:**
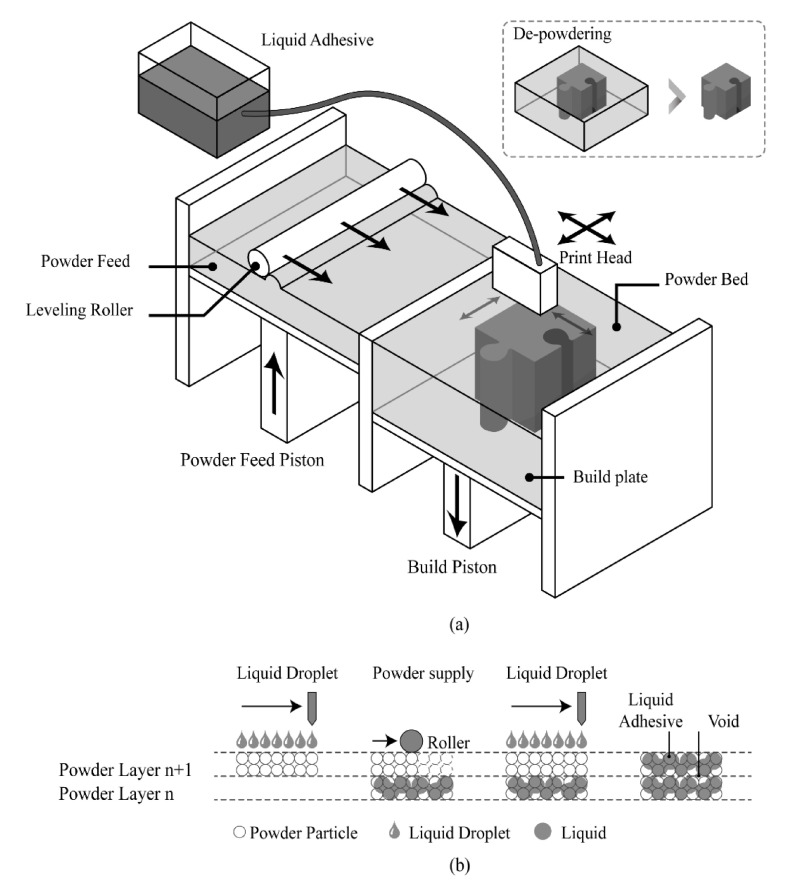
Schematic diagram of binder jetting 3D printing technology: (**a**) the BJ system; (**b**) powder/binder interaction between adjacent layers [44].

**Figure 8 sensors-24-02668-f008:**
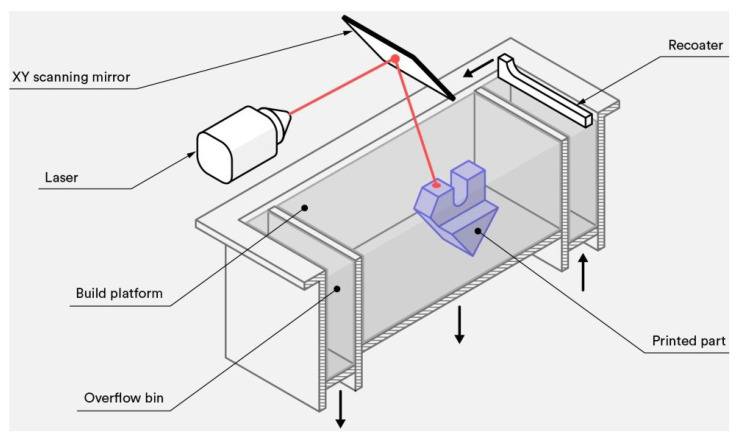
Schematic of SLS 3D printers [55].

**Figure 9 sensors-24-02668-f009:**
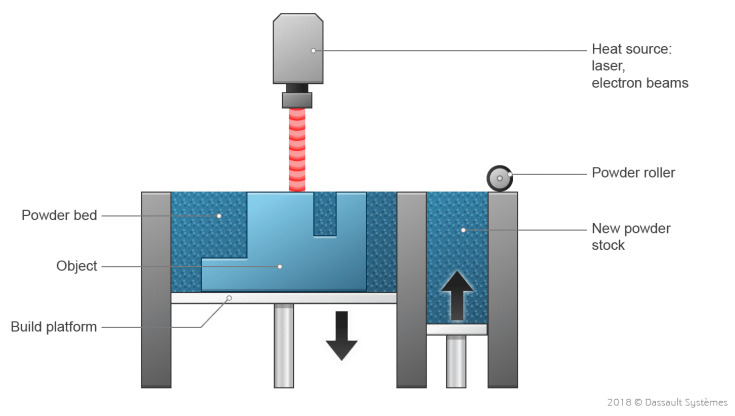
Schematic of powder bed fusion [65].

**Figure 10 sensors-24-02668-f010:**
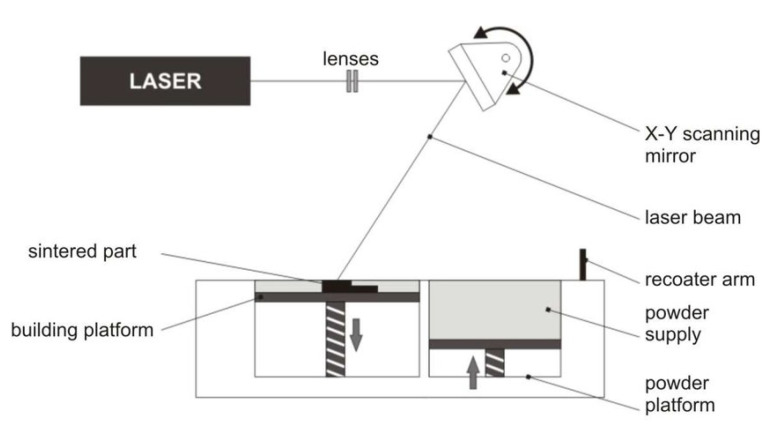
Schematic of direct metal laser sintering [87].

**Figure 11 sensors-24-02668-f011:**
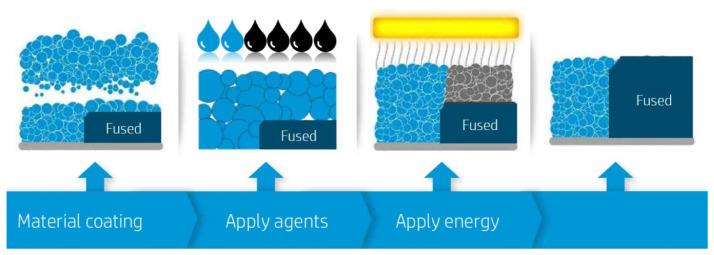
Schematic diagram of multi-jet fusion (MJF) [103].

**Figure 12 sensors-24-02668-f012:**
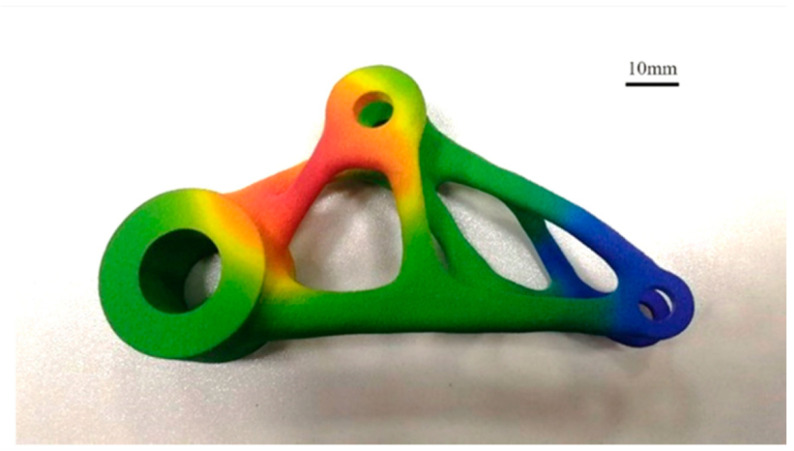
Sample 3D-printed part through the MJF technology by HP [106].

**Figure 13 sensors-24-02668-f013:**
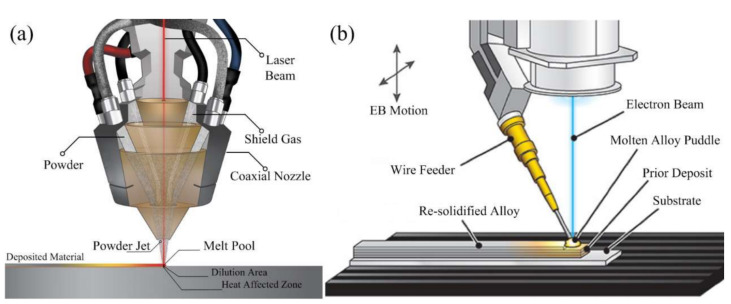
Schematic of directed energy deposition (DED): (**a**) Powder DED (laser source); (**b**) Wire DED (E-beam source) [111].

**Figure 14 sensors-24-02668-f014:**
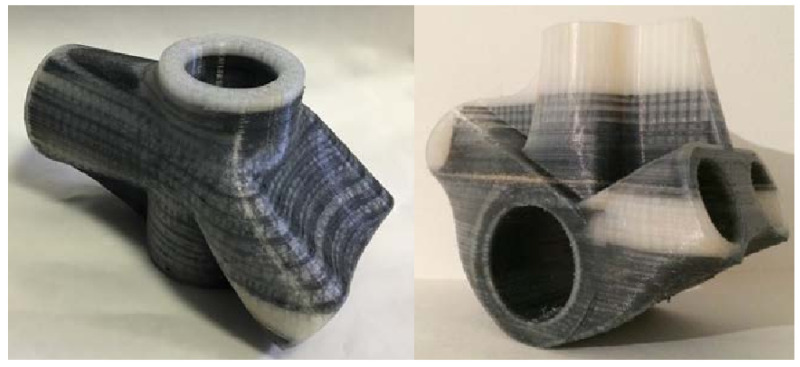
Three-dimensionally-printed bicycle lugs reinforced with continuous carbon fiber [117].

**Figure 15 sensors-24-02668-f015:**
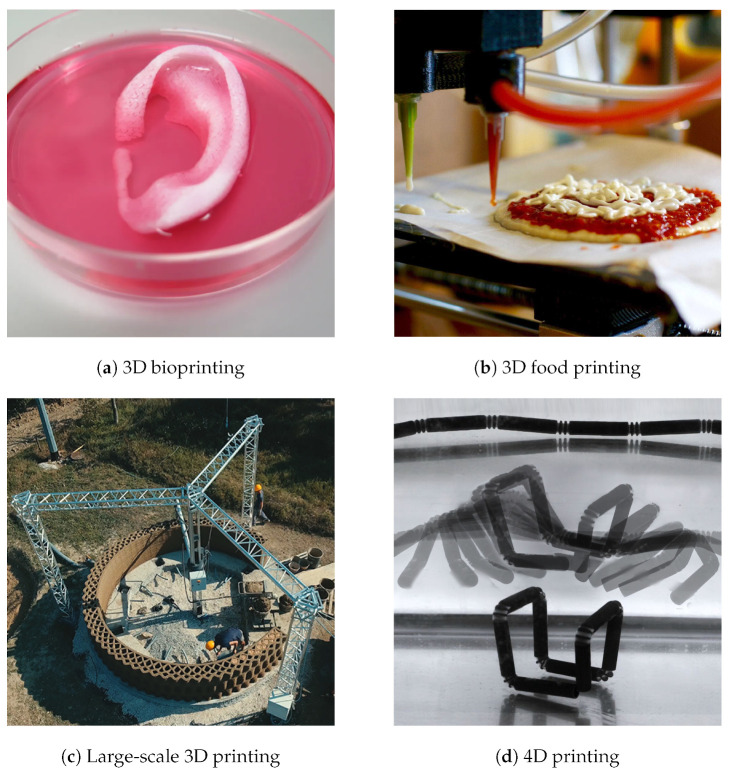
Developing trends of additive manufacturing technologies: (**a**) a 3D-bioprinted human tissue (an ear) with biodegradable plastic scaffolding using an integrated tissue and organ printing system (ITOP) by the Wake Forest Institute for Regenerative Medicine [124]; (**b**) a 3D-printed pizza by Beehex through a fund from NASA [125]; (**c**) the Crane Wasp 3D printer, created to print homes using local materials [126]; (**d**) a ’smart’ self-folding material that can transform shape, printed by MIT’s Self-Assembly Lab [127].

**Table 1 sensors-24-02668-t001:** AM technology comparison.

AM Tech.	Advantages	Limitations	Common Materials
FDM	Low price Speed efficiency Low maintenance	Requires high temperatures Requires supports Inability for certain geometries	PLA ABS PETG
VP	Exceptional details and surface Ideal for intricate features	Limited build volume Shrinkage and warping Toxicity and environmental concerns	Photocurable resins Waxes Ceramics
MJ	Exceptional resolution and details Multi-material and full-color Wide array of materials	High costs Restricted build volume Low printing speed	Photopolymers Thermoplastic polymers Metal powders
BJ	Full-color Various materials	Low-density parts Labor-intensive Complex post-processing	PVP PVA PAA
SLS	Support free Isotropic final products Complex geometries	Porosity, shrinkage, and impurities Poor surface quality Post-processing for final appearance	Plastics Composites Ceramics
SLM	Binder-free Often faster than SLS High powder recyclability	High costs Less material flexibility Support structures and inert gas	Titanium alloys Stainless steels Aluminum alloys
DMLS	Support-free Various materials of metal alloy High power recyclability	High costs High porosity Limited build volume	Stainless steels Aluminum Titanium
EBM	Excellent material properties Processing reactive metals Fast and efficient process	High equipment costs Additional post-processing Limited selection of materials	Titanium alloys Nickel-based superalloys Cobalt–chrome alloys
MJF	High speed and efficiency Less post-processing Multiple colors High ductility	Material limitations Specific use cases	Polyamides Thermoplastic Polyurethanes Polypropylene
DED	Minimizes waste Part repair and modification Efficiency for larger components	Post-processing for a smooth finish Low precision	Stainless steel Titanium alloys Nickel-based alloys
CFR	Strength and lightness Customizable fiber orientation	High costs Limited material compatibility	Nylon + Carbon fiber PEEK + Carbon fiber ABS + Carbon fiber
LOM	Cost-effectiveness and high speed Environmental friendliness Large part creation	Low dimensional accuracy and strength Post-processing for surface finish Limited material range for composites	Paper Metal foils Plastics

*Note*: A list of abbreviations is in the Abbreviations section at the end of the manuscript.

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
