# Peer review of "Additive Manufacturing: A Comprehensive Review"

_sensors, 2024, doi:10.3390/s24092668_

Round 1
Reviewer 1 Report
Comments and Suggestions for Authors
The authors provided a systematic introduction for additive manufacturing that is of great significance especially in the field of high-value added industry. The manuscript is in good written. I would like to recommend acceptance with the following points being addressed:
- Lack of references in Introduction section for the purpose of supporting the authors’ opinion.
- Line 111, missing number of figure (perhaps Fig. 1?)
- In the section of Additive Manufacturing Process, the abbreviation “AM” appeared, and it reappeared in section 2.1. Please recheck the whole manuscript ensure the abbreviations are properly used. I suggest a list of nomenclatures in the beginning.
- Section 3.1, Line 243: The authors mentioned “All 3D printing involves three main steps, pre-processing, production, and post-processing.” Please specify the meaning of pre-processing. E.g.: Is it utilized for material only or for machines? To my knowledge, although the pre-processing for materials is always preferred for higher quality, it is not necessary for all kinds of additive manufacturing machines. This topic was nether discussed in the rest manuscript.
- In section 3.6.3, thermal crack should be included as a bottleneck in the field of SLM.
- What is the intention of reintroducing DMLS after the part of SLM?
- In section 3.8.3, the general requirement for EBM precursor powder should be added, as well as the fabricating precision. In metallic AM, the consideration of the limitation from powder is one of the most critical step. I also encourage a similar discussion in the section of DED.
Author Response
Dear Reviewer, Thank you very much for your time in evaluating our manuscript titled “Additive Manufacturing: A Comprehensive Review”. We carefully considered all your comments and revised the manuscript accordingly. For your convenience, we listed all reviewers’ comments and our responses (in red color) one by one in the attached "Revision Report". We also highlighted all revisions in the manuscript. We hope the issues have been addressed in this revision. Feel free to let us know if any other questions. Thank you!

Reviewer 2 Report
Comments and Suggestions for Authors
The article covers in detail all the aspects of various AM process. Except for few grammatical and spell checks rest of the article seems ok. However, I do not see any novelty in this review with respect to discussion of various processes which is carried out by many authors. Try to include some new case studies and comparative tables in various sections.
Comments on the Quality of English Language
Moderate corrections are required
Author Response

(The authors gave the same response as above.)

Reviewer 3 Report
Comments and Suggestions for Authors
The paper requires more in-depth analysis to improve its quality. A comparison table is suggested by showing pros and cons, compatible materials, applications, costing, environmental impact, etc for each technology. Detail criteria for comparison for the technologies are required. For the application, it is suggested to add the quantitative or qualitative results. Section 4.3 to 4.5 need more in-depth analysis, for e.g., case studies shall be included and discussed. Quantitative potential can be added, for e.g., LCA analysis etc. The novelty of the paper is weak. Line 1139 - incomplete sentence. Line 1019-incorrect format.
Author Response

(The authors gave the same response as above.)

Reviewer 4 Report
Comments and Suggestions for Authors
This work fully summarizes the process of 3D printing technology, the advantages and disadvantages of various types of 3D printing technology and the performance of printing, but some of the problems need to be modified more often.
1. As a review article, the author lacks the expression of pictures in this research, such as the process principles of various processes, and specific applications in those places, these need specific pictures to express, which may be more conducive to readers to understand various processing methods.
2. The author introduces a lot of processing technology, illustrates the use of materials, applications, etc., also describes the advantages and disadvantages of various kinds of processes, but the lack of a visual horizontal contrast, Suggestions in the third part to add a horizontal contrast table, the printable material types, processing resolution, processing size size do a horizontal contrast.
3. The title of this article is “Additive Manufacturing: A Comprehensive Review”. However, all the technologies reviewed in this paper are macro-scale 3D printing technologies, and they do not summarize the technology and process of microscale 3D printing, which is inconsistent with the title of this paper. It is suggested to add the relevant parts of microscale 3D printing and consider adding more to the relevant literature, such as aerosol jet printing(,DOI: 10.34133/research.0164, DO: I10.1002/adem.202100362, DO: I10.1016/j.sna.2023.114777) Electrohydrodynamic jet printing(DOI: 10.34133/cbsystems.0091, DOI: 10.1149/1945-7111/ab9c7e, DOI: 10.1002/aelm.202200728), laser direct writing(DOI: 10.1088/2631-7990/ab0edc, DOI: 10.1088/2631-7990/acf798) electric field-driven jet printing(DOI: 10.1002/adma.202007772), etc.
4. The fourth part only explains the application trend of the future development but does not explain the direction of the development from the perspective of technology and process. It is suggested to add this part.
Author Response

(The authors gave the same response as above.)

Reviewer 5 Report
Comments and Suggestions for Authors
Dear authors,
The paper "Additive Manufacturing: A Comprehensive Review" analyzes the additive manufacturing (AM) field, emphasizing its revolutionary role across various industries by facilitating the creation of complex geometries and custom designs and optimizing material usage. It covers AM's evolution from a rapid prototyping instrument to an industrial manufacturing method utilizing diverse materials like polymers, metals, ceramics, and composites. The review discusses the entire AM workflow, including design, modeling, printing, and post-processing. It also examines various AM technologies, such as material extrusion, selective laser sintering, and direct metal laser sintering. AM's prospects include advancements in 3D bio-printing, large-scale printing, 4D printing, and AI integration.
While the paper is a valuable resource for understanding the scope of additive manufacturing, addressing the comments stated by the reviewer and general areas for improvement will significantly enhance its contribution to the academic community. These suggestions will provide an improved perspective on AM's current state, challenges, and future directions, improving the paper substantially.
General Comments
The paper offers an overview of additive manufacturing technologies and their transformative impact on different industries/areas. The exploration of AM workflows and future advancements provides valuable insights. However, to enhance the paper's value and relevance, specific areas require improvement:
- A detailed comparison of the mechanical performance limitations of AM structures with those produced by traditional manufacturing methods would improve the analysis. The paper might benefit from a more detailed comparison of AM technologies against conventional manufacturing processes, highlighting specific use-case advantages.
- Incorporating more empirical data from existing studies and case studies can strengthen argumentation and provide practical insights into the application of AM technologies across different industries. This would offer readers a clearer understanding of the benefits and limitations of AM in real-world settings. Integrating examples of AM success in optimizing industrial processes could illustrate the technology's practical benefits.
- A general economic analysis of the additive manufacturing process could be valuable, including cost-benefit analysis, production efficiency, and return on investment for different AM technologies. This would help readers understand the financial implications of adopting AM processes.
- Briefly expanding the introduction on the environmental impact of AM technologies, including energy consumption, material waste, and the lifecycle analysis of AM-produced parts, could provide a more complete view of the sustainability of these technologies.
- The sections and subsection structure of the paper could be improved for readability.
- Adding figures and tables summarizing the key points and comparing technologies could enhance the paper's clarity and benefit readers.
- Addressing standardization and regulatory challenges in additive manufacturing could provide valuable insights into the obstacles facing the broader adoption of AM technologies by industry and consumers.
- Although it discusses AM's challenges, the paper needs a detailed analysis of solutions or ongoing research to overcome them.
Specific Comments
- Line 33: Replace "no-double" with "no doubt" to correct the typographical error.
- Line 11: A missing figure number should be included for clarity.
- Line 252: Discuss high-performance thermoplastics like PEEK or Ultem, acknowledging their requirement for higher temperatures.
- Section 3.1.5: Mention filament core reinforcement using glass/carbon/basalt fibers and nanotechnology trends in FDM.
- Section 3.2.5: Elaborate on the biocompatibility of VAT process materials.
- Section 4.2: Discuss limitations in optimizing the rheological properties of printing products.
- Section 4.3: Highlight challenges in structural reinforcement integration and the need for regulation and standardization.
- Line 1258: Include shape memory alloys as smart materials in this context
- Conclusions Enhancement: The conclusions section requires further development to highlight emerging trends. What is the authors' take on the significant development potential in AM?
Author Response

(The authors gave the same response as above.)

Round 2
Reviewer 2 Report
Comments and Suggestions for Authors
Review article is modified.
Author Response
Thank you again for your time in reviewing our paper. We really appreciate your effort and contributions in the process.

Reviewer 3 Report
Comments and Suggestions for Authors
The revised version is greatly improved. It is suggested to use passive voice.
Author Response
Thank you. We have revised to passive voice as much as we can.
Reviewer 4 Report
Comments and Suggestions for Authors
I agree that this article is published in its present form, although I still believe that the authors need to add more pictures for characterization. The current pictures are not enough to reflect 3D printing in a more intuitive way. I think there is still a lack of process and specific equipment pictures to intuitively express the 3D printing process.
Comments on the Quality of English LanguageA small amount of English editing is required.
Author Response
Thank you again for your time in reviewing our paper. We have added more pictures to enhance the visualization

Reviewer 5 Report
Comments and Suggestions for Authors
The authors have made a significant effort to improve the manuscript. I appreciate the commitment to improving the quality of the manuscript. The manuscript has been significantly improved. Therefore, I suggest that the paper be published as is.
Author Response
Thank you again for your time and effort in reviewing our paper. We really appreciate your effort and contributions in the process.